# Sources of error in open-path FTIR measurements of $N_2O$ and $CO_2$ emitted from agricultural fields

Cheng-Hsien Lin[1], Richard H. Grant[1], Albert J. Heber[2], and Cliff T. Johnston[1,3]

[1]Department of Agronomy, Purdue University, West Lafayette, IN 47907, United States
[2]Department of Agricultural and Biological Engineering, Purdue University, West Lafayette, IN 47907, United States
3Department of Earth, Atmospheric and Planetary Sciences, Purdue University, West Lafayette, IN 47907

*Correspondence to*: Cheng-Hsien Lin (lin471@purdue.edu)

**Abstract.** Open-path Fourier transform infrared spectroscopy (OP-FTIR) is susceptible to environmental variables which can become sources of errors for gas quantification. In this study, we assessed the effects of water vapour, temperature, path length,
and wind speed on quantitative uncertainties of nitrous oxide ($N_2O$) and carbon dioxide ($CO_2$) derived from OP-FTIR spectra. The presence of water vapour in spectra underestimated $N_2O$ mole fractions by 3 % and 12 %, respectively, from both lab and field experiments using a classical least squares (CLS) model when the reference and sample spectra were collected at the same temperature (i.e. 30 °C). Differences in temperature between sample and reference spectra also underestimated $N_2O$ mole fractions due to temperature broadening and the increased interferences of water vapour in spectra of wet samples. Changes in
path length resulted in a non-linear response of spectra and bias (e.g. $N_2O$ and $CO_2$ mole fractions were underestimated by 30 % and 7.5 %, respectively, at the optical path of 100 m using CLS models). For $N_2O$ quantification, partial least squares (PLS) models were less sensitive to water vapour, temperature, and path length, and provided more accurate estimations than CLS. Uncertainties in the path-averaged mole fractions increased in low wind conditions ($< 2$ m s$^{-1}$). This study identified the most common interferences that affect OP-FTIR measurements of $N_2O$ and $CO_2$, which can serve as a quality assurance/control
guide for current or future OP-FTIR users.

## 1 Introduction

Agriculture substantially contributes greenhouse gases (GHGs), mostly $CO_2$, $N_2O$, and $CH_4$, to the atmosphere (IPCC, 2007). In 2010, emissions led by agricultural activities (e.g. crop production and livestock management) were estimated between 5.2 and 5.8 Gt of $CO_2$ equivalent per year, accounting for 10-12 % global anthropogenic emissions (IPCC, 2014). Gas flux
estimations over an extended period (e.g. growing seasons) are complicated due to the dynamic and episodic nature of gas emissions and measurement complexities. Integrated uses of fast-response gas concentration sensors and micrometeorological techniques were developed to continuously measure the long-term gas fluxes (Baldocchi, 2003; Denmead, 2008; Flesch et al., 2016). Open-path Fourier transform spectroscopy (OP-FTIR) is capable of measuring multiple gases simultaneously with high temporal and spatial resolution through a real-time measurement and the path-averaged concentration (Russwurm and Childers,
2002). OP-FTIR has been applied to measure GHGs, and other trace gases (e.g. $NH_3$) emitted from agricultural fields (Childers

et al., 2001a and 2001b; Bjorneberg et al., 2009; Flesch et al., 2016; Lam et al., 2017). Using OP-FTIR to quantify gas concentrations, however, is a complicated process. Each step in data collection and spectral analyses influences the accuracy and precision of gas quantifications, including spectral resolution, apodization function, choice of background (e.g. zero-path or synthetic backgrounds), and chemometric models (Russwurm and Childers, 1999; Griffiths and de Haseth, 2007; Hart and
Griffiths, 2000; Hart et al., 2000). Also, OP-FTIR spectra are sensitive to ambient environmental conditions (e.g. humidity, air temperature, optical distance, and wind speed) which interfere with spectral analyses and quantification of gas concentrations (Griffiths and de Haseth, 2007; Muller et al., 1999; Shao et al., 2007 and 2010).

Water vapour is the major interfering species in FTIR-derived trace gas quantification due to its strong absorption features
within the mid-infrared region (400-4000 cm$^{-1}$) (Russwurm and Childers, 1999; ASTM, 2013). For interference-free spectra, a single absorption line can be easily isolated to calculate gas concentrations (i.e. univariate methods). Using this method, however, is challenging to adequately isolate the absorption feature of the gas of interest from water vapour (Muller et al., 1999; Briz et al., 2007). Multivariate methods have been proposed to improve gas quantification by selecting broad spectral windows of gases of interest (Haaland and Easterling, 1980; Haaland and Thomas, 1988; Hart and Griffiths, 1998 and 2000;
Hart et al., 1999 and 2000; Muller et al., 1999). The most common method in commercial quantification packages is the classical least squares (CLS) quantitative model (Griffiths and de Haseth, 2007). Studies showed that the interference of water vapour was mitigated by either considering reference spectra of water vapour or through selection of appropriate spectral windows in CLS models (ASTM, 2013; Haaland and Easterling, 1980; Hart and Griffiths, 2000; Horrocks et al., 2001; Jiang et al., 2002; Du et al., 2004; Briz et al., 2007; Lin et al., 2019). Other studies, however, found that CLS models resulted in a
substantial error for quantifying the targeted gas under the interference from the non-targeted gases (mostly water vapour) even if the reference spectra of all gas species and the optimal spectral window were considered (Hart et al., 1999; Briz et al., 2007; Shao et al., 2010; Lin et al., 2019). As a result, the partial least squares (PLS) algorithm was proposed to improve the accuracy of gas quantification (Haaland and Thomas, 1988; Hart et al., 2000; Shao et al., 2010; Lin et al., 2019).

The features of gas rotation-vibrational absorption bands are temperature-dependent (Lacome et al., 1984; Rothman et al., 2005). Ideally, sample and reference spectra should be collected at the same temperature to avoid temperature-related bias (Russwurm and Childers, 1999; ASTM, 2013). Training spectra for building quantitative models, such as CLS, were generally collected at the same temperature. The non-linear responses of spectral absorbance to significant diurnal variations in temperature leads to errors in gas quantification (Russwurm and Phillips, 1999; de Castro et al., 2007; Smith et al., 2011). For
continuous field measurements, it is time-consuming to create piecewise models to cover entire ranges of 1) the path-integrated concentrations of gases of interest and interferences, and 2) temperature. A PLS model was developed to cover wide ranges of environmental variables, including concentrations, path lengths, humidity, and temperature (Bjorneberg et al., 2009; Griffiths et al., 2009; Shao et al., 2010 and 2011).

In addition to changes in water vapour content and temperature, the experimental configuration and optical parameters also influence OP-FTIR spectra. Compared with extractive methods, one of the advantages of OP-FTIR is to use different path lengths to measure gases from multiple sources (Russwurm and Childers, 1999; Bacsik et al., 2006). Short optical path lengths can over saturate the detector that introduces error in gas quantification (Bartoli et al., 1974; Chase, 1984; Griffiths and de Haseth, 2007). A long path length increases quantitative sensitivities, but the increased length reduces the intensity of incident signals and decreases the signal-to-noise (S/N) ratio (Griffith and Jamie, 2006; Nelson et al., 1996; Kosterev et al., 2008). Spectral resolution and apodization also affect the spectral linearity (Griffiths, 1994; Zhu and Griffiths, 1998; Russwurm and Phillips, 1999; Childers et al., 2002). Lower resolution spectra are incapable of resolving absorption features. Even though the apodized interferogram (IFG) can reduce noises (or spurious oscillations) of a single-beam (SB) spectrum converted by Fourier transformation, different apodization functions affect spectrum linearity differently. The non-linear relationship between absorbance and concentrations disobeys Beer-Lambert Law and leads to biases in gas quantification (Haaland, 1987; Russwurm and Phillips, 1999; Childers et al., 2001a).

The CLS algorithm is subject to the nonlinearity induced by ambient interferences (e.g. multiple overlapping components) or instrumentation setup (e.g. low spectral resolution) which can be overcome by the PLS-based quantitative model (Hart et al., 2000). Alternatively, the non-linear least squares (NLLS) regression function was also proposed to correct non-linear behaviour of the spectrum to concentrations and improve the quantitative accuracy (Smith et al., 2011; Griffith et al, 2012; Paton-Walsh et al., 2014; Phillips et al., 2019). The NLLS coupled with a forward modelling approach, such as the Multi-Atmospheric Layer Transmission (MALT) method developed by Griffith (1996) or the E-trans software developed by Ontar Corporation, can model a transmission spectrum from the HITRAN database by selecting specific parameters (including temperature, pressure, path length, resolution, and apodization function) to fit the sampled SB spectrum for gas quantification. One of the benefits of the MALT is that a 'background' SB spectrum (the spectra without gases of interest) is not required in the procedure for quantification, which can avoid the background-induced errors (e.g. zero-path background or the zapped background shown in Lin et al, 2019). Nevertheless, it is challenging to consider other variable sources in the field (e.g. severe spectral overlap or wind-induced variations) using the MALT which is built upon the HITRAN library. Instead, the PLS-based approach can be trained empirically and consider the local environmental variables in quantitative models for field gas measurements (ASTM, 2013; Griffiths et al., 2009).

Many OP-FTIR field studies used either CLS or PLS as frameworks for gas measurements, and most of these studies minimized environmental interferences (e.g. water vapour or wind speed) by developing methods for spectral analyses and gas quantification (Hong and Cho, 2003; Hart et al., 1999 and 2000; Muller et al., 1999; Childers et al., 2002; Briz et al., 2007; Shao et al., 2007; Griffiths et al, 2009; Shao et al., 2010; Lin et al., 2019). Briz et al. (2007) examined the difference in water vapour content (0.5 % vs. 2.5 %) on trace gas quantification but their study did not include $N_2O$. Shao et al. (2007) investigated the effect of wind on spectrometer vibrations and spectra qualities but did not address its influence on gas quantification.

Temperature effect on spectral parameters (e.g. absorption intensity and bandwidth) of single gas components have been well established (e.g. high-resolution transmission molecular absorption, HITRAN, database published by Rothman et al., 2005); however, only limited studies assessed influences of dynamic temperature on gas quantification using OP-FTIR. Horrocks et al. (2001) and Smith et al. (2011) used OP-FTIR spectrometer and a gas cell purged with dry samples to measure the temperature-related error in $SO_2$, CO, $CO_2$, and $CH_4$ quantification, but not for $N_2O$.

To the best of our knowledge, there has never been a study of exploring the influence of changing temperature and path length on $N_2O$ quantification under water vapour interferences in spectra. To test the quality of OP-FTIR methods for multiple gas quantification is challenging due to interferences and the lack of proper measurement benchmarks. Also, the influence of water vapour is confounded by temperature and path length effects. In this study, the influence of water vapour, temperature, path length, and wind speed on $N_2O$ and $CO_2$ quantification are examined using lab and field-based (OP-FTIR) methods.

## 2 Materials and experimental methods

### 2.1 Lab FTIR experiment

The lab FTIR experiment was used to assess the effects of water vapour and air temperature on $N_2O$ quantification from spectra. An FTIR spectrometer equipped with a variable-path length gas cell was used to acquire reference spectra of $N_2O$, water vapour, and $N_2O$ plus water vapour mixtures (i.e. wet $N_2O$) as shown in Fig. 1a.

#### 2.1.1 Instrumentation setup

The lab FTIR spectrometer (Nexus 670, Thermo Electron Corporation, Madison, WI) was equipped with a KBr beam splitter, and a high-D* MCT detector was used to analyse gas samples using a multi-pass gas cell (White cell) (model MARS-8L/40L, Gemini Scientific Instruments, CA) with an optical path length of 33 m. Spectra range of 700-4000 $cm^{-1}$, optical velocity of 0.6 cm $s^{-1}$, and a resolution of 0.5 $cm^{-1}$ were selected for spectra acquisition. Each spectrum was acquired by co-adding 64 IFGs using the OMNIC software package (Thermo Fisher Scientific, Inc.) A triangular function was applied to apodize an IFG for converting an SB spectrum using the Fourier transform. A sampled SB spectrum contained the visible information of gases of interest and interferences. A background SB spectrum was collected from pure $N_2$ and used to convert a sampled SB spectrum to an absorbance spectrum. The temperature of the gas cell was controlled and monitored (model 689-0005, Barnant Co., Barrington, IL). The spectrometer was purged with dry air (−20 °C dew point) from a zero-air generator (model 701H, Teledyne, Thousand Oaks, CA). Gas samples were produced using a diluter (series 4040, Environics Inc, Tolland, CT) with the accuracy of ±1.0 % for mole fraction measurements. The mole fraction (ppbv) of $N_2O$ was diluted with ultra-pure $N_2$ gas. Mole fractions of water vapour (ppmv) were controlled by a Nafion tube (Perma Pure, Lakewood, NJ) enclosed within a sealed container containing saturated water vapour. The saturated water vapour content in the container was adjusted by controlling temperature of the water bath. Wet $N_2O$ gas samples were produced by passing dry $N_2O$ from the diluter through the Nafion

tube with a constant flow rate (4 L·min$^{-1}$). Gas samples were continuously introduced into the White cell with a constant pressure close to the room ambient pressure. Humidity and temperature probes (model HMT330, Vaisala Oyj, Helsinki, Finland) were used to monitor the humidity and temperature of the introduced gas samples. In this study, $N_2O$ (dry and wet) was diluted from 30 ppmv ($N_2O$ in $N_2$) to 310, 400, 500, 600, and 700 ppbv with various water vapour contents (a relative
humidity of 20 %, 40 %, 60 %, and 80 % at 30 °C).

### 2.1.2 Data collections and gas quantification

Spectra were collected when the concentrations of $N_2O$ and water vapour and temperature of the introduced samples were steady. A total of nine single-gas absorption spectra, five dry $N_2O$ spectra with mole fractions of 310-700 ppbv, and four water vapour spectra with mole fractions of 7000-28 000 ppmv, were used to train CLS models (CSL-1 shown in Table 1). A total
of sixty mixed-gas (wet $N_2O$) spectra were used to build PLS models (see the table S2a in the supplement). Spectral windows (Table 2) and linear baseline correction were applied in CLS-1 and PLS models to calculate mole fractions of $N_2O$ and water vapour from the mixed-gas spectra (the validation/wet sample spectra shown in Table 1) using TQ Analyst software Version 8.0 (Thermo Fisher Scientific, Inc.) The previous study showed that the integrated window of 2188.7-2204.1 cm$^{-1}$ and 2215.8-2223.7 cm$^{-1}$ ($W_N3$) is less sensitive to changing environment (e.g. water vapour = 5000-20 000 ppmv; ambient temperature =
10-35 °C) and provides higher accuracy than the window of 2170.0-2223.7 cm$^{-1}$ ($W_N1$) for $N_2O$ quantification using either CLS or PLS models (Lin et al., 2019). These two windows ($W_N1$ and $W_N3$) were used to calculate $N_2O$ mole fractions and examine the effect of water vapour, temperature, and path length on $N_2O$ calculations in this study. In PLS models, optimum factors were determined by cross-validation and justified by the prediction of residual error sum of squares (PRESS) function to avoid over-fitting issues (see table S2b in the supplement). Since it was difficult to isolate the effects of the temperature and
humidity on quantitative biases from the field experiment, the validation spectra with the fixed mole fractions of $N_2O$ and water vapour (310 ppbv $N_2O$ mixed with 21 500 ppmv water vapour) collected at 30, 35, and 40 °C from the lab-FTIR to evaluate the sensitivity of CLS and PLS to temperature.

### 2.2 Open-path FTIR experiment

The OP-FTIR experiment was used to assess the effects of water vapour, air temperature, path lengths, and wind speed on the
quantification of $N_2O$ and $CO_2$ from field spectra. The field instrumentation and configuration were shown in Fig. 1b.

### 2.2.1 Site description and instrumentation setup

The field study was conducted at Purdue University Agronomy Center for Research and Education of West Lafayette, Indiana, the United States (86°56´ W, 40°49´ N). The predominant soil series at the study site was a Drummer silty clay loam (fine-silty, mixed, mesic Typic Endoaquoll). The bulk density of topsoil (0-10 cm) and organic matter (0-20 cm) was measured 1.6
g·cm$^{-3}$ and 3.8 %, respectively. The experimental site (Purdue Field Trace Gas Flux Facility) was between two maize cropping fields (~10 m apart) with anhydrous ammonia applications. A monostatic OP-FTIR (IR source, interferometer, transmitting

and receiving telescope, and detector combined in one instrument) spectrometer (model 2501-C, MIDAC Corporation, Irvine, CA) was used to collect field-IR spectra. A retroreflector with 26 corner-cubes (cube-length of 76 mm) was mounted on a tripod 150 m from the spectrometer corresponding to an optical path of 300 m. The experiment of varying path lengths was conducted using optical path lengths of 100, 200, and 300 m.

## 2.2.2 Data collections and gas quantification

The same sample collection parameters were used to collect both OP-FTIR and lab-FTIR spectra. OP-FTIR spectra were collected using the AutoQuant Pro4.0 software package (MIDAC Corporation, Irvine, CA). Each field spectrum was collected by co-adding 64 IFGs and a resolution of 0.5 $cm^{-1}$. Triangular apodization, Mertz phase correction, and no zero-filling (i.e. spectral data point spacing ~ 0.25 $cm^{-1}$) were applied to convert an IFG into an SB spectrum. A stray-light SB spectrum was collected every thirty minutes by pointing the spectrometer telescope away from the retroreflector and subtracted from sampled SB spectra for stray-light correction. Quality control and assurance procedures (Russwurm, 1999; ASTM, 2013; Russwurm and Childers, 1999; Childers et al., 2001b; Shao et al., 2007) were used to evaluate spectra qualities and the influence of wind-induced vibrations. Gas mole fractions derived from the OP-FTIR spectra were also calculated by CLS and PLS models. Training absorption spectra used in the CLS model were generated from the HITRAN database using E-trans (Ontar Corporation North Andover, MA). Briefly, high-resolution spectral lines of $N_2O$, $CO_2$, and water vapour output from E-trans were interpolated to generate spectra ranging from 500 $cm^{-1}$ to 4000 $cm^{-1}$ and convolved with a triangular apodization function. The convolved spectral lines were used to generate the reference spectra with the identical resolution and data point density matching the field spectra using Grams/32 (Childers et al., 2001). The HITRAN reference spectra were generated at the pressure of 760 torr and the temperature of 30 °C.

The stray-light corrected field SB spectrum was converted to absorbance spectra by the synthetic SB background (syn-bkg) spectra using the IMACC Quantify package (Industrial Monitoring and Control Corp., Round Rock, TX). The syn-bkg was generated by selecting multiple points from the spectral interval of interest (i.e. six points within 2050–2500 $cm^{-1}$ for $N_2O$ and $CO_2$) to fit the curvature of the sample SB spectrum using a polynomial function (Lin et al., 2019). Three spectral windows (Table 2) and the HITRAN references were used to build the CLS model (CLS-2 shown in Table 1) in the IMACC software. The third-degree polynomial function was used to correct the non-linear response of the CLS calculations to the actual mole fraction. More details regarding the IMACC quantification package were described in the IMACC user manual attached as the supplementary material. Since the IMACC software only allowed users to build the CLS model, the PLS model was also built based on the lab-FTIR reference spectra using the TQ Analyst software for $N_2O$ estimations from the OP-FTIR spectra (Table 1). Since the molar fraction changes with changing air density which is the function of temperature and pressure, the measured temperature and pressure in the gas cell and experimental field were imported in the quantitative software to adjust the model-calculated mole fractions.

Ambient temperature, relative humidity, and barometric pressure in the field were measured using an HMP45C probe (Vaisala Oyj, Helsinki, Finland) and a pressure sensor (278, Setra, Inc., Boxborough, MA) at 1.5 m above ground level (a.g.l.). The mean wind speed was measured by a 3-D sonic anemometer (model 81000, RM Young Inc., Traverse City, MI) mounted at 2.5 m a.g.l. and recorded at 16 Hz. A 50-m synthetic open path gas sampling system (S-OPS) (Heber et al., 2006) was used to collect gas samples along the OP-FTIR optical path to analyse the path-averaged concentrations of $N_2O$ and $CO_2$ using a difference frequency generation mid-IR (DFG-IR) laser-based $N_2O$ gas analyser (IRIS 4600, Thermo Fisher Scientific Inc., Waltham, MA) and a non-dispersive IR (ND-IR) $CO_2$ gas analyser (LI-840, LI-COR Inc., Lincoln, NE), respectively. The $N_2O$ and $CO_2$ analysers provided a high precision for $N_2O$ (< 0.15 ppbv, 1σ) and $CO_2$ (< 1.0 ppmv, 1σ) measurements, and both analysers were calibrated using the certified standard gas (±1 % accuracy) based on the US-EPA protocol every four hours to ensure the instrumentation stability and measurement accuracy. Mole fractions of $N_2O$ and $CO_2$ were measured from both S-OPS and OP-FTIR simultaneously, and the S-OPS measurements were used as benchmarks to examine the accuracy and the sensitivity of OP-FTIR on gas quantification (Lin et al., 2019). The surface layer of air tends to become homogeneous in a well mixing condition (i.e. wind speed > 1.7 m s$^{-1}$ shown in Lin et al., 2019), and the well-mixed atmospheric condition ideally minimizes spatial variations in the path-averaged concentrations from different measurement units (i.e., 50-m S-OPS vs. 150-m OP-FTIR).

### 2.2.3 Path lengths experiment

A variable path length between an OP-FTIR spectrometer and a retroreflector resulted in different path-integrated mole fractions (e.g. ppm-m) as well as the depth of gas absorbance in SB spectra. The complexities of $N_2O$ absorption features within the 2170-2224 cm$^{-1}$ range required high spectral resolution (Fig. 3). For $N_2O$, the increased absorbance, resulting from a longer path length, likely improves its quantitative sensitivity and accuracy. Spectra were collected from physical lengths of 50, 100, and 150 m (Fig. 1b) using the same parameters. During the measurement (14:30-18:30, local time (LT) on 6 May 2016), the background mole fractions of $N_2O$ (349.0 ± 0.5 ppbv) and $CO_2$ (400.0 ± 4.4 ppmv) which were determined from the S-OPS, ambient temperature and humidity (the relative humidity of 35 % at 25 °C) remained nearly constant. The spectra acquired from different path lengths were analysed by CLS-2 models to calculate mole fractions of $N_2O$ and $CO_2$ and by PLS models only for $N_2O$. In a well-mixed air condition, the path-averaged concentrations measured from the 50-m S-OPS can be also used to assess the quantitative accuracy from different path lengths (i.e. 100- and 150-m OP-FTIR).

### 2.4 Quantitative accuracy

Quantitative accuracy/bias was determined by the relative error between the FTIR calculations ($x_i$) and the true gas mole fractions ($x_t$) of either the introduced gas (Lab) or the S-OPS measurements (Field), following Eq. (1):

$$\text{Bias} = [(x_i - x_t)/x_t] \times 100 \,\% \tag{1}$$

## 3 Results and discussion

Accurate gas quantification from either lab- or field-based FTIR spectra requires knowledge of the optimum spectral window (the spectral region used for quantification). In general, broadening spectral windows will contain more spectral features that can be useful for quantification. At the same time, however, broader windows will also contain more contributions from interfering constituents (e.g. water vapour). The optimum window would have clean spectral features of the target species with minimal spectral interference from other gases. For $N_2O$ quantification, our previous work showed that the optimum window was to integrate two intervals of 2215.8-2223.7 and 2188.5-2204.1 $cm^{-1}$ ($W_N3$ shown in Lin et al., 2019). In this paper, two windows ($W_N1$ and $W_N3$) and models (CLS and PLS) from the previous study were used to predict $N_2O$ mole fractions.

### 3.1 Lab FTIR experiment

### 3.1.1 Water vapour effect

Water vapour interfered with spectral windows and resulted in underestimations of $N_2O$ mole fractions using CLS models; increased water vapour increased the bias (Fig. 2). The PLS model provided more accurate predictions for $N_2O$ than the CLS. The CLS accurately predicted gas concentrations only when the water vapour was absent or limited in spectra (Hong and Cho, 2003; Esler et al., 2000; Shao et al., 2010; Smith et al., 2011). In open-path measurements, CLS was often observed to underestimate gas concentrations, as reported by Childers et al. (2002), Briz et al. (2007), Shao et al. (2010), and Lin et al. (2019). Absorbance spectra of dry and wet $N_2O$ showed that water vapour interferences likely compromised the intensity of the $N_2O$ P-branch absorbance (Fig. 3). Ideally, the absorbance/intensity of the 310 ppbv $N_2O$ should be identical in either dry or wet conditions. The wet spectrum (red solid line), however, showed lower $N_2O$ intensity than the dry spectrum (black solid line) after the baseline correction (Fig. 3). The reduced $N_2O$ absorbance in wet samples resulted in underestimations of $N_2O$ mole fractions using CLS models that were created based on references of dry $N_2O$ samples and water vapour. PLS models, created by wet $N_2O$ references, showed improved accuracy in wet samples but overestimated $N_2O$ in dry samples (Fig. 2). It is still unclear how water vapour interfered with gas quantification. The $N_2O$ underestimation (based on the CLS model predictions) resulting from the attenuated absorbance was hypothesized due to the inadequate spectral resolution. High resolution is required to resolve rotation-vibrational gas spectral features (e.g. full-width at half height ~ 0.2 $cm^{-1}$) to avoid spectral nonlinearity to concentrations (ASTM, 2013; Griffiths and de Haseth, 2007; Russwurm and Phillip, 1999; Muller et al., 1999). Absorption features of $N_2O$ were strongly overlapped by water vapour within 2170-2224 $cm^{-1}$. In order to resolve absorbance spectra of multiple gases and spectral overlaps, spectral resolution higher than 0.2 $cm^{-1}$ would be suggested. Increased optimal resolution, however, is a trade-off for the ratio of signal to noise which is along with detection limits as well as quantitative precision.

### 3.1.2 Temperature effect

The temperature-sensitivity of gas-phase FTIR spectra results in non-linearity of absorbance to temperature. Bias will be introduced if there is a temperature difference between reference and sample spectra (Russwurm and Phillip, 1999; Smith et al., 2011). The effect of this delta temperature on $N_2O$ quantification is shown in Fig. 4a. Spectra of wet $N_2O$ (310 ppbv $N_2O$ blended with 21 000 ppmv water vapour) were collected at 30 °C, 35 °C, and 40 °C. Reference spectra of dry $N_2O$, water vapour, and wet $N_2O$ were acquired at 30 °C and used to calculate $N_2O$ mole fractions from spectra collected at temperatures of 35 °C and 40 °C. The difference in temperature led to biases in $N_2O$ calculations (Fig. 4a). Smith et al. (2011) calculated concentrations of $CO_2$, $CH_4$, and CO using the MALT (Griffith, 1996) and showed that temperature-related error was approximately 3 % when the delta temperature was within 10 °C. As mentioned, water vapour present in spectra resulted in $N_2O$ underestimations using CLS models (Fig. 2), and this bias further increased with increasing delta temperature. For instance, the bias increased from -3 % to -5 % with increasing temperature from 30 °C to 40 °C using CLS models (Fig. 4a). Sources to this bias appeared to include 1) temperature-broadening of $N_2O$ and 2) temperature-induced interference of water vapour (i.e. water vapour interference increased with increasing temperature). Increased temperature tends to broaden the width of absorption lines in the rotation-vibration $N_2O$ spectra and results in underestimations of gas mole fractions. Increased water vapour strength in spectra led to more interferences in spectral analyses and biases (Fig. 4b). Even though it is challenging to avoid interferences from water vapour, a proper window selection (e.g. $W_N1$ vs. $W_N3$) can mitigate the effect of water vapour on gas quantification. PLS methods also showed more accurate estimations and less sensitivity to temperature than CLS models (Fig. 4a).

### 3.2 Open-path FTIR experiment

### 3.2.1 Water vapour effect

In fields, water vapour ranged from 5000 to 20 000 ppmv during 9-19 June 2014. Increased water vapour showed the increased $N_2O$ biases using CLS (Fig. 5a and 5b). In the lab experiment, water vapour increasing from 5000 to 20 000 ppmv at 30 °C only showed consistent underestimations for $N_2O$ quantification by approximately 3 % using CLS (Fig. 2). The positive relationship between water vapour content and the $N_2O$ bias (Fig. 5a) was because water vapour was confounded by temperature. Increased temperature (i.e. 10-35 °C in fields) tends to increase humidity in the air. This negligible correlation ($R^2 = 0.20$ shown in Fig. 5a) became insignificant when the calculated biases were categorized by temperature (e.g. $R^2 = 0.01$ at the interval of 25-30 °C, data not shown). $CO_2$ measured by CLS was less sensitive to water vapour content than $N_2O$ in field measurements ($R^2 = 0.05$ shown in Fig. 5b), presumably due to the less water vapour absorption features in 2075.5-2084.0 $cm^{-1}$ than 2170.0-2224.0 $cm^{-1}$ (Lin et al., 2019). For PLS calculations, $N_2O$ biases became consistent but slightly increased with increasing water vapour (Fig. 5c).

### 3.2.2 Temperature effect

Increased air temperature increased both $N_2O$ and $CO_2$ biases from CLS models (Fig. 5d and 5e). A strong correlation of air temperature to $N_2O$ biases ($R^2 = 0.86$) showed that $N_2O$ quantification was more sensitive to temperature effects than $CO_2$ ($R^2 = 0.39$). The lab experiment (Fig. 4) showed that CLS underestimated $N_2O$ by 3 % in wet air for the samples with a low delta-

temperature. $N_2O$ calculations from OP-FTIR spectra, however, were underestimated by 12 % (approximately 36 ppbv less than the true value) even if the HITRAN reference and sample spectra were collected at the same temperature (i.e. 30 °C). The excess bias (12 % minus 3 %) likely resulted from interferences from CO and $CO_2$ in 2170-2224 cm$^{-1}$ and inherent uncertainties in line intensities and bandwidths of gas absorbance from HITRAN database (Rothman et al., 2005). The CLS-calculated $CO_2$ values were less influenced by temperature than $N_2O$ (Fig. 5e), attributed to the reduced complexity of $CO_2$ absorption features

in the 2075-2085 cm$^{-1}$ region, and less interference of water vapour within this region (Lin et al., 2019). Since the temperature-dependent absorption lines vary with species and wavelengths, the resolution parameter and gas quantification react differently to a changing environmental temperature (Smith et al., 2011; Griffith et al., 2012).

The PLS model were not only less affected by temperature ($R^2 = 0.05$) but provided better accuracy for $N_2O$ estimations (Fig.

5f). For instance, $N_2O$ bias was reduced from -12 % (the CLS-calculated $N_2O$ shown in Fig. 5d) to approximately 2 % using the PLS model at 30 °C (Fig. 5f). This slightly overestimated $N_2O$ mole fraction (i.e. 2.2±0.8 % shown in Fig. 5f) was possibly due to 1) the limited information regarding other sources of environmental variables (e.g. wind-driven variables such as the homogeneity of the mixing air) in the PLS model, and 2) different background gas compositions between training and field spectra. The pure $N_2$ was used as a buffer gas to generate reference mixtures (i.e. $N_2O$ mixed with water vapour) to train the

PLS model. For field spectra, however, only approximately 78 % $N_2$ is present in the spectra. Since the line-broadening coefficient is sensitive to the background compositions (Lacome et al., 1984), the discrepancy in backgrounds (i.e. pure $N_2$ vs. air) likely leads to errors in gas quantification. Furthermore, the linewidth of absorption bands is influenced by both temperature and pressure broadening effects (Loos et al., 2015). Variations in pressure between the reference and sample spectra also reduce the quantitative accuracy (Smith et al., 2011). In the control environment experiment, the pressure effect

on bandwidth (or the associated error) was minimized due to the nearly constant pressure in the gas cell. The ambient air pressure was measured from 981 to 996 hPa during the measurement period. Compared with the wide ranges of temperature and water vapour content, the effect of the small variations in the pressure (989.3±3.2 hPa, n = 355) on gas quantification was not examined, and the measured pressure was only used to adjust the model-calculated mole fractions in this study.

### 3.2.3 Path length effect

OP-FTIR spectra containing constant mole fractions of $N_2O$ and $CO_2$ (i.e. $N_2O$ ~ 349.0 ± 0.5 ppbv and $CO_2$ ~ 400.0 ± 4.4 ppmv) were collected from optical lengths of 100, 200, and 300 m. As path lengths decreased, both $N_2O$ and $CO_2$ were underestimated (Fig. 6a and 6b). For $N_2O$, CLS calculations were more sensitive to path lengths than PLS (Fig. 6a). The Beer-

Lambert law should show a linear response of absorbance to the path-integrated concentration. Nevertheless, the path-averaged absorbance of $N_2O$ and $CO_2$ (i.e. $\frac{Absorbance}{Path\ length(m)}$) did not conform to the Beer-Lambert law even though ambient mole fractions of $N_2O$ and $CO_2$ were nearly consistent (Fig. 6c and 6d), showing that there was a non-linear response of OP-FTIR spectra to the path-integrated concentrations. Several reasons may have caused non-linearity issues, such as detector saturation, spectral resolution, and apodization (ASTM, 2013; Russwurm and Childers, 1999; Griffiths and de Haseth, 2007). Detector saturation at short distances was avoided in this study by examining the IFG centre burst and SB spectra. For instance, the spectra were excluded if either the maximum or minimum signal of the centre burst exceeded the detector A/DC capacity (~ 2.5 V). Also, the elevated baseline below the detector cut-off, usually 600 cm$^{-1}$, in the SB spectrum was used as an indicator to inform the detector saturation (ASTM, 2013).

Thus, this short-path-derived bias (Fig. 6a and 6b) likely resulted from the inadequate spectral resolution. Short path lengths reduced the absorbance depth in an SB spectrum as well as sensitivity for quantification. Poorly resolved absorbance spectra could lead to a spectral non-linear response with different path-integrated concentrations (Zhu and Griffiths, 1994; Russwurm and Phillips, 1999). Also, $N_2O$ quantification was more sensitive to path lengths than $CO_2$. By increasing optical path lengths from 100 m to 300 m, the accuracy of $N_2O$ calculated from CLS models increased by approximately 20 % ($N_2O$ biases reduced from −30 % to −10 % shown in Fig. 6a). For $CO_2$, the accuracy only increased by 2.5 % ($CO_2$ biases reduced from −7.5 % to −5.0 % shown in Fig. 6b). The difference in sensitivity between gas quantification and path length was attributed to the complexity of absorbance spectra. $N_2O$ absorption features in 2170-2224 cm$^{-1}$ were more complicated than $CO_2$ in 2075-2085 cm$^{-1}$; furthermore, more interfering gases (CO, $CO_2$, and water vapour in 2170-2224 cm$^{-1}$) interfered with $N_2O$ quantification (Lin et al., 2019). A triangular apodization function applied in spectra results in a non-linear response (Russwurm and Phillips, 1999). The poorly-resolved spectra containing multiple gas species likely complicated the magnitude of the non-linearity led by apodization, which, however, was not evaluated in this study. The PLS methods reduced $N_2O$ biases and showed less sensitivity to path length than CLS (Fig. 6a).

### 3.2.4 Wind speed effect

Nitrous oxide is predominately produced via soil microbial activities (nitrification and denitrification), and $CO_2$ is from respirations of soil microbes and vegetations (Mosier et al., 1996). As a result of soil and crop heterogeneities, multiple sources, and intermittent fluxes of $N_2O$ and $CO_2$ from the soil and/or canopy result in inhomogeneous gas concentrations in the atmosphere under low winds. Since the path length of the S-OPS (50-m) was different from the OP-FTIR (the physical length of 150 m), the gas uniformity across the 150 m influenced their path-averaged concentrations. The difference in the path-averaged $N_2O$ and $CO_2$ mole fractions between the S-OPS and OP-FTIR was used to calculate quantification bias (Fig. 7). Variations in $N_2O$ and $CO_2$ biases were small but increased when the wind speed was less than 2 m s$^{-1}$ (Fig. 7c and Table S3 in the supplement). This increased variability inferred the poorly mixed air (< 2 m s$^{-1}$). Thus, decreasing wind speed and

turbulent mixing tended to increase the differences in gas concentration between the S-OPS and OP-FTIR. During the low wind environment, $CO_2$ bias showed higher variability than $N_2O$, presumably due to a greater environmental variation in $CO_2$ concentrations than $N_2O$ (Lin et al., 2019). For instance, $CO_2$ can be produced from both soil and canopy respiration, and plant uptake via photosynthesis. $N_2O$ was predominately produced from soil nitrification and denitrification. The $CO_2$ concentrations and their spatial distribution in the air were influenced by the variations in both soil properties and crop species (different sources). Thus, $CO_2$ concentrations in the atmosphere tended to have higher variations than $N_2O$ and become highly heterogeneous if the air was poorly mixed in the low wind condition. Low winds likely occurred during the night period. During 9-19$^{th}$ 2014, a total of 259 and 130 data points (30-min averages $N_2O$) were collected from the daytime (06:00-20:00, LT) and nighttime (20:00-06:00, LT) measurements, respectively. The low wind condition can occur during both day and night even though low winds are more common during the night. In this study, 22% (57 out of 259) of all daytime measurements and 36% (47 out of 130) of all nighttime measurements were collected from the low wind environment ($< 2$ m s$^{-1}$) (Fig. 7a).

## 4 Conclusion and recommendations

In this study, we have evaluated the effects of water vapour, temperature, path length, and wind speed on open-path FTIR measurements of $N_2O$ and $CO_2$ quantified using CLS and PLS models. Water vapour existing in spectra underestimated $N_2O$ mole fractions by 3 % (lab experiment) and 12 % (field experiment) at 30 °C using CLS models. PLS models improved the accuracy of $N_2O$ quantification (lab bias = $-0.6 \pm 0.4$ % and field bias = $2.0 \pm 0.8$ %). Differences in temperature between reference and sample spectra led to errors in gas quantification. Increased air temperature significantly increased quantification bias using CLS models. For wet $N_2O$, 10 °C difference introduced 2 % (Lab) and 9 % (Field) more biases in gas mole fractions. PLS models were less sensitive to temperature. Short path lengths reduced the sensitivity and accuracy for gas quantification, and CLS models were more sensitive to the changed path lengths than PLS. These short-path-led biases were presumably due to the inadequate spectral resolution. $CO_2$ quantification using CLS model was less influenced by environmental variables than $N_2O$ likely due to its less complex absorption features. The wind affected the mixings of gases and the low wind speed ($< 2$ m s$^{-1}$) led uncertainties in the path-averaged mole fractions.

The PLS model generally provided more accurate measurements than CLS if the gas of interest is strongly interfered with by water vapour/other interfering gases (e.g. strong overlap of water vapour absorbance features or broad spectral windows). PLS is also less sensitive to environmental variables than CLS. For OP-FTIR measurements, the CLS-calculated concentrations need to be verified carefully for quality assurance and to avoid substantial underestimations. Path lengths must be adequate, which can be checked by conducting a path length experiment before measurements. For the users interested in multi-source measurements, we suggested avoiding a great difference in path lengths if CLS models are used for gas quantification. High spectral resolution ($< 0.5$ cm$^{-1}$) is recommended to resolve complex spectral features of either gas of interest or interferences. A high resolution also introduces more noises and increasing the scan time is suggested to increase the signal-to-noise ratio (Griffiths and de Haseth, 2007).

*Author contributions.* CHL, CTJ, RHG, and AJH designed the lab- and field-FTIR measurement experiments. CHL and RHG conducted the field experiment. CHL conducted the lab experiment, spectral and data analyses, and prepared the manuscript with contributions from CTJ, RHG, and AJH.

*Competing interests.* The authors declare that they have no conflict of interest.

*Acknowledgements.* The authors appreciate the crop and field management from Tony Vyn and Terry West, technical assistance from Austin Pearson and Allison Smalley, additional travel funding support from the Purdue University Climate

Change Research Center, and the technical support from the United States Environmental Protection Agency (US-EPA).

*Financial support.* This work was funded by the United States Department of Agriculture National Institute for Food and Agriculture, USDA NIFA (grant no. 13-68002-20421), and the Indiana Corn Marketing Council (grant no. 12076053).

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

**Table1.** Multivariate models (i.e. CLS and PLS) used for $N_2O$ and $CO_2$ quantification: 1) the CLS-1 model was built by five dry $N_2O$ spectra (310, 400, 500, 600, 700 ppbv) and four water vapour spectra (7 000, 15 000, 22 000, 28 000 ppmv) collected from the lab-FTIR, 2) CLS-2 model was built by twelve dry $N_2O$, seventeen dry $CO_2$, and sixteen water vapour spectra generated from the HITRAN database (see Table S1 in the supplement), and 3) the PLS model was built by a total of sixty wet $N_2O$ spectra collected from Lab-FTIR (see Table S2 in the supplement).

| Model | Training/Reference spectra | Source | Validation/Wet Sample spectra | Source |
|-------|---------------------------|--------|-------------------------------|--------|
| CLS-1 | Dry $N_2O$: 310-700 ppbv<br>Water vaour: 7,000-28,000 ppmv | Lab-FTIR | $N_2O$ | Lab-FTIR |
| CLS-2 | Dry $N_2O$, $CO_2$, water vapour: see Table S1 | HITRAN | $N_2O$, $CO_2$ | OP-FTIR |
| PLS | Wet $N_2O$: see Table S2 | Lab-FTIR | $N_2O$ | Lab- and OP-FTIR |

**Table 2.** Spectral windows for quantification of $N_2O$ and $CO_2$.

| Gas | Windows (cm$^{-1}$) | Interferences |
|-----|---------------------|---------------|
| $N_2O$ | $W_N1$: 2170.0-2223.7 | $H_2O$, CO, $CO_2$ |
| | $W_N3$: 2188.7-2204.1 + 2215.8-2223.7 | $H_2O$, CO, $CO_2$ |
| $CO_2$ | $W_C2$: 2075.5-2084.0 | $H_2O$ |

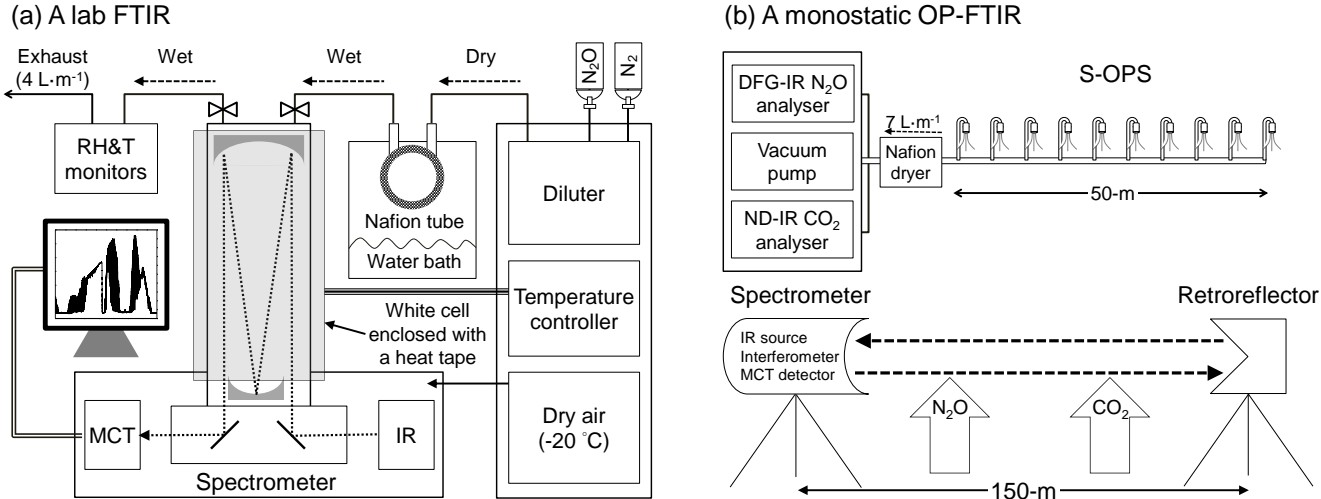

**Figure 1.** Schematic of the instrumentation used to assess the effects of water vapour and temperature on gas quantification: (a) lab-FTIR with a multi-pass gas cell (optical path length of 33 m); (b) DFG-IR $N_2O$ and ND-IR $CO_2$ analysers combined with a synthetic open path gas sampling system (S-OPS) were used as benchmarks to assess quantification of $N_2O$ and $CO_2$ from OP-FTIR.

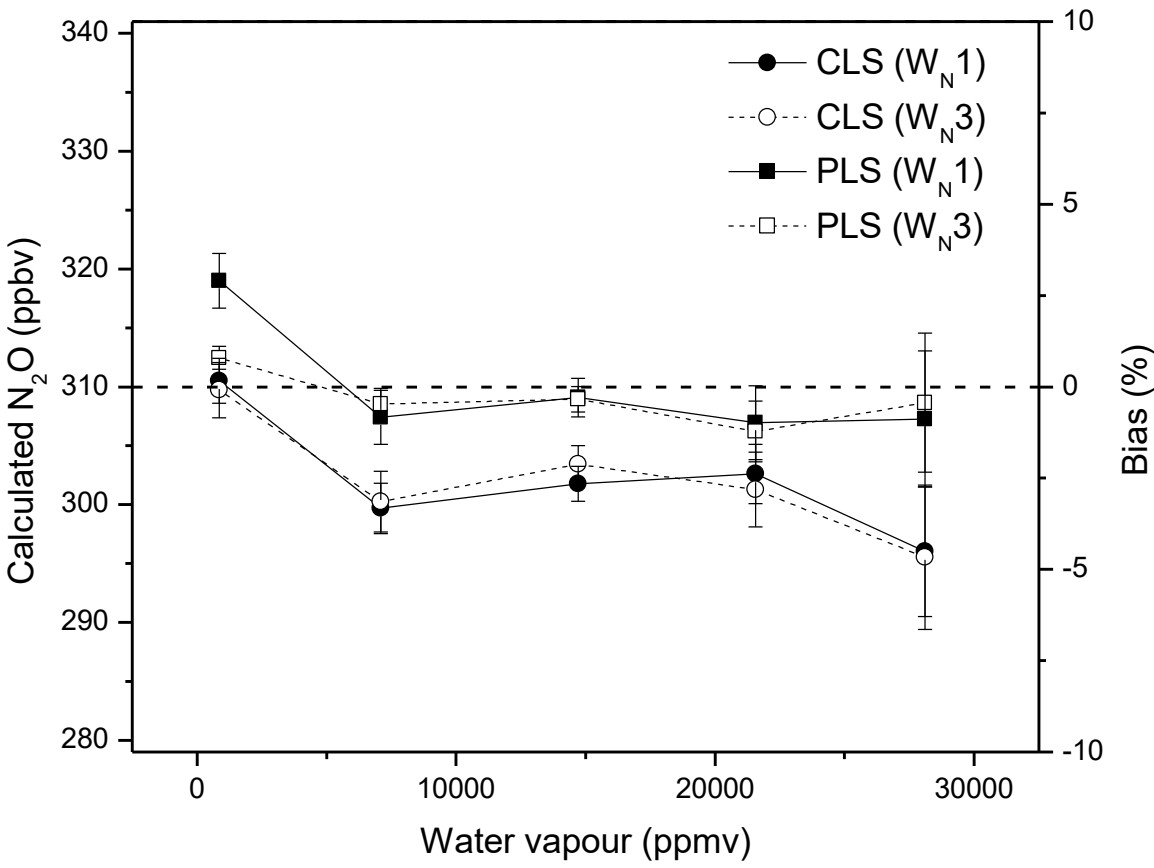

**Figure 2.** Effects of water vapour on $N_2O$ quantification: the lab-FTIR spectra of dry $N_2O$ and $N_2O$/water vapour mixtures (310 ppbv $N_2O$ at the relative humidity of 20 %, 40 %, 60 %, and 80 % at 30 °C) were used to calculate $N_2O$ mole fractions using CLS and PLS models and two spectral windows ($W_N1$ and $W_N3$).

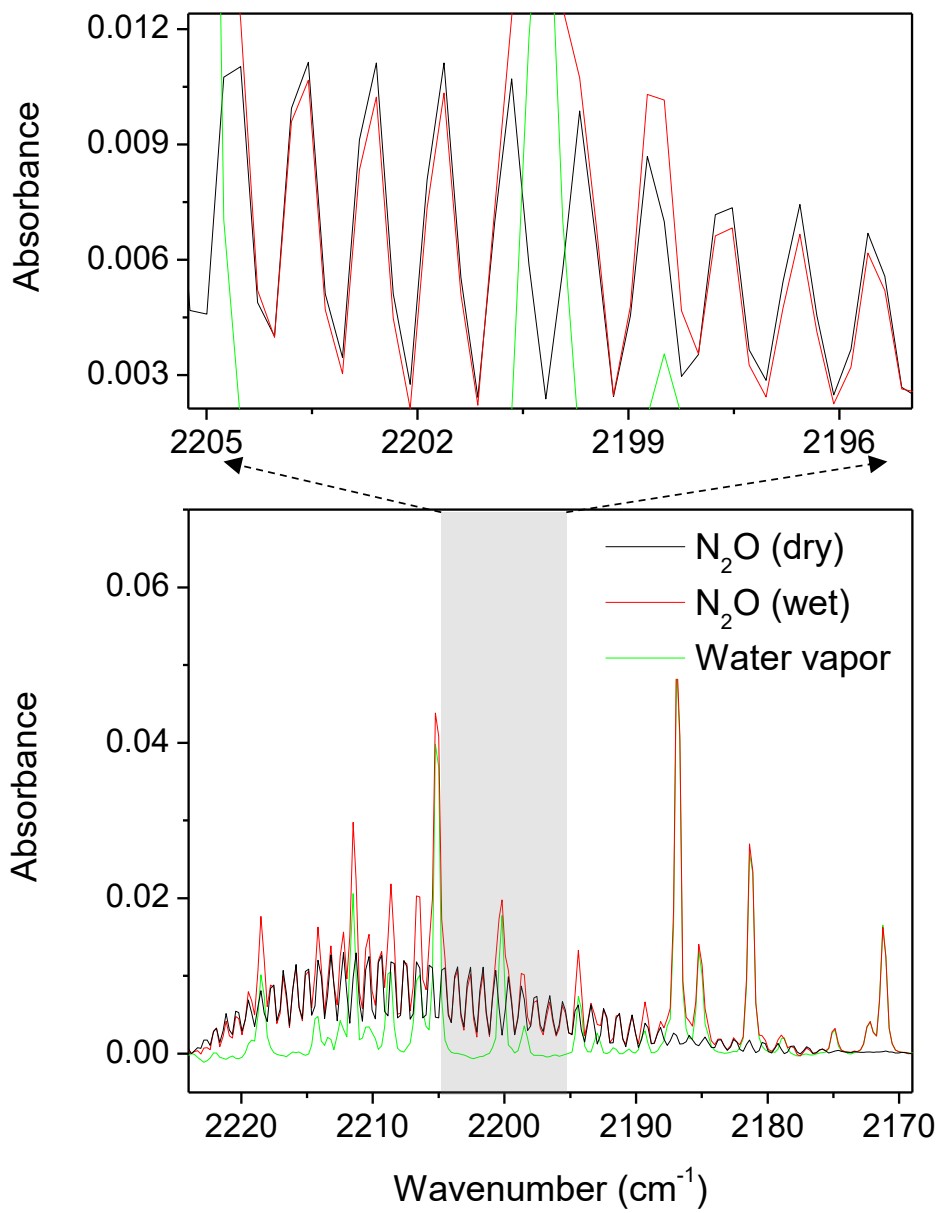

**Figure 3.** The lab-FTIR spectra of dry $N_2O$ (310 ppbv), wet $N_2O$ (310 ppbv $N_2O$ plus 28 000 ppmv water vapour), and water vapour (28 000 ppmv) were acquired at 30 °C.

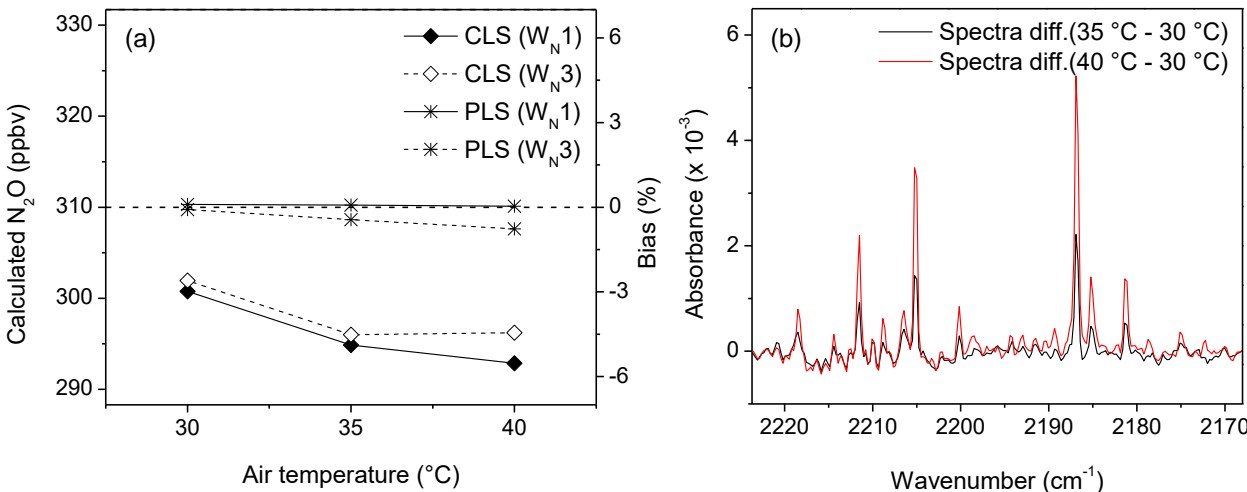

**Figure 4.** Effects of temperature on $N_2O$ quantification: the lab FTIR spectra containing wet $N_2O$ (310 ppbv $N_2O$ plus 21 500 ppmv water vapour) were acquired at 30 °C, 35 °C, and 40 °C. Temperature affected (a) $N_2O$ mole fractions calculated by CLS and PLS models and two spectral windows ($W_N1$ and $W_N3$) and (b) spectral differences in wet $N_2O$ absorbance (310 ppbv $N_2O$ plus 21 500 ppmv water vapour) among different temperature.

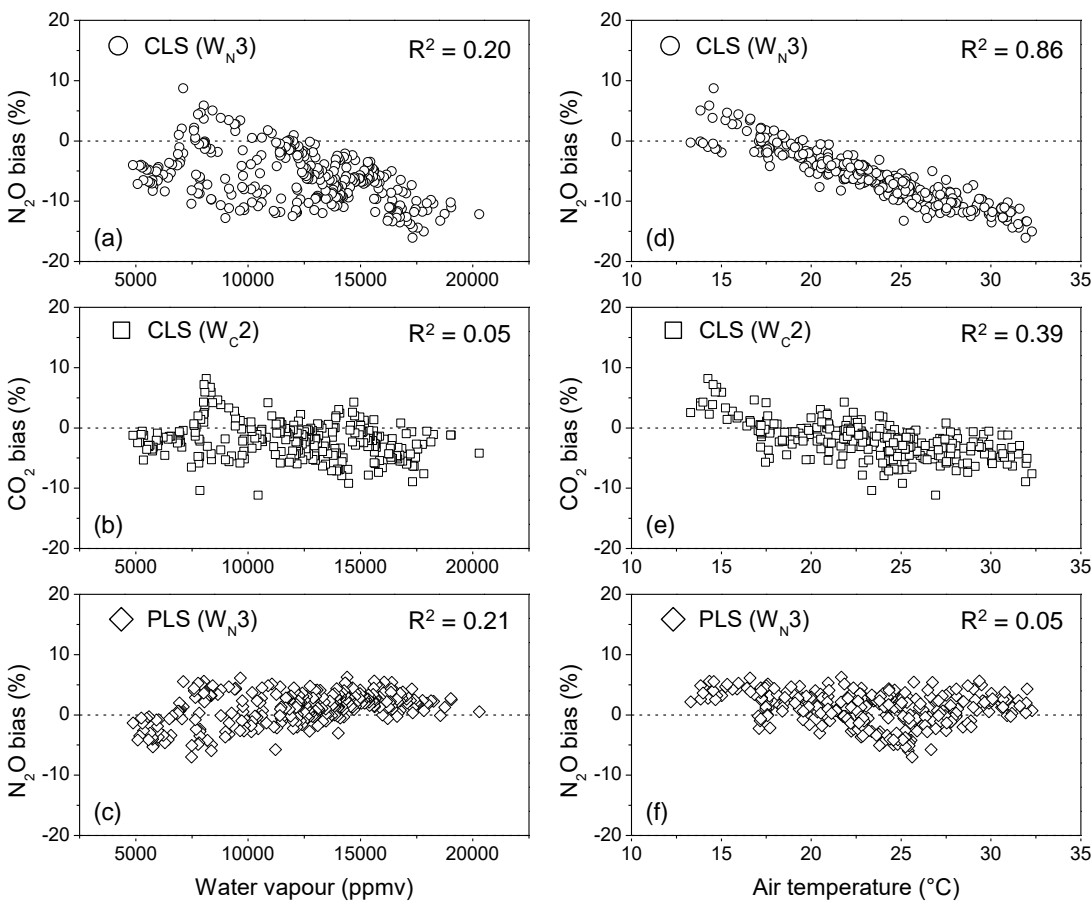

**Figure 5.** Effects of water vapour and temperature on N$_2$O and CO$_2$ quantification from the OP-FTIR spectra using CLS and PLS models and the optimum windows (W$_N$3 for N$_2$O and W$_C$2 for CO$_2$) during 9-19 June 2014. Assumed temperature and bias are a linear relationship.

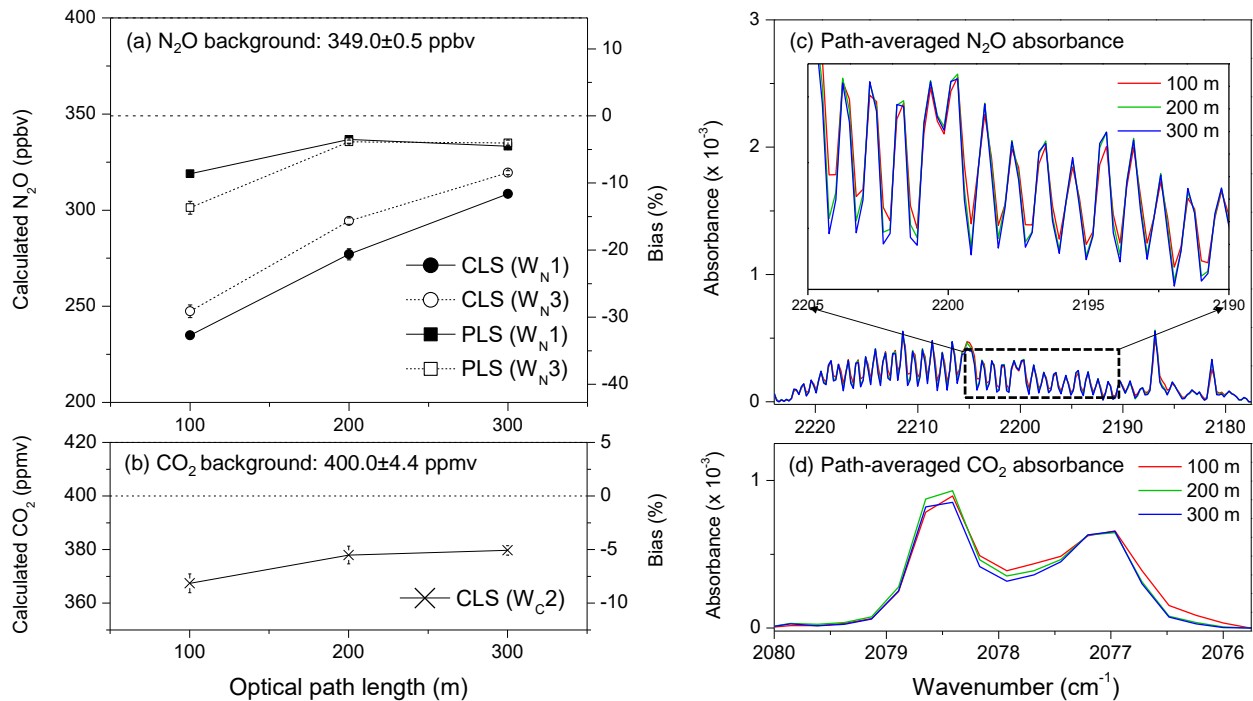

**Figure 6.** Effects of path lengths on $N_2O$ and $CO_2$ quantification: OP-FTIR spectra were acquired under backgrounds of $N_2O$ (349.0 ± 0.5 ppbv) and $CO_2$ (400.0 ± 4.4 ppmv), and relative humidity of 35 % at 25 °C from optical path lengths of 100, 200, and 300 m for quantification of (a) $N_2O$ using CLS and PLS and windows of $W_N1$ and $W_N3$, (b) $CO_2$ using a CLS model and the window of $W_C2$. The path-averaged absorbance spectra of (c) $N_2O$ and (d) $CO_2$ showed the inconsistency of absorbance spectra.

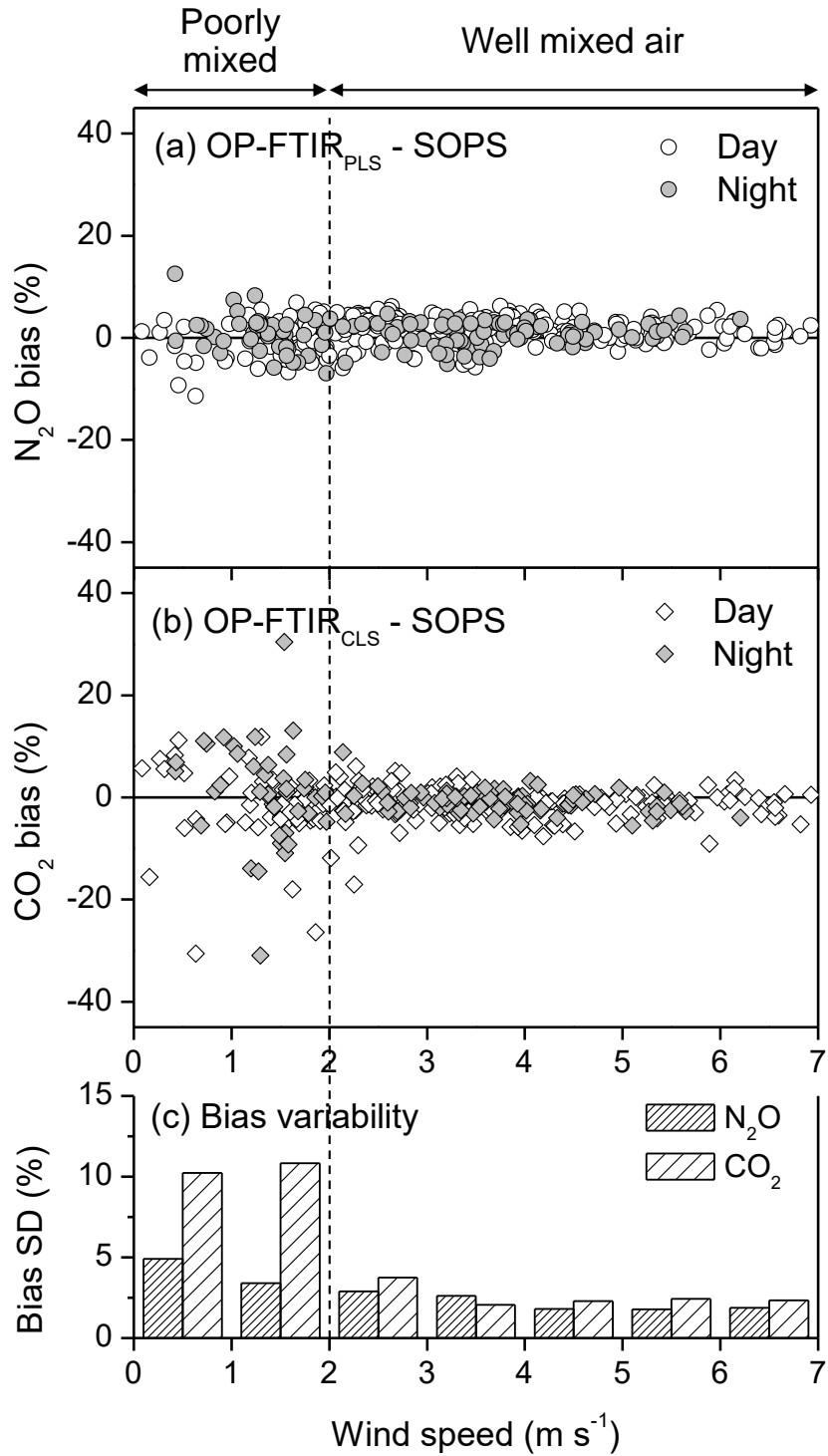

**Figure 7.** Effects of the wind speed on differences in the path-averaged concentrations between the OP-FTIR and S-OPS: (a) $N_2O$, (b) $CO_2$, and (c) variability of biases (Standard deviation, SD).