# Peer review of "Sources of error in open-path FTIR measurements of $N_2O$ and $CO_2$ emitted from agricultural fields"

_Atmospheric Measurement Techniques, 2019_

## Referee Comment (RC1) · Anonymous Referee #1 · 21 Aug 2019

I found this to be one of the most useful papers that I have read on open-path FT-IR spectroscopy. I have only a few comments.

Both Griffiths and Russwurm have shown that triangular apodization causes greater deviations from Beer's law non-than other apodization functions and so it is a pity that this was the apodization function used in this study. Incidentally, rather than citing the short paper written by Zhu and Griffiths given in this paper, either of the following two citations would have been better: "Errors in Absorbance Measurements in Infrared Fourier Transform Spectrometry Because of Limited Instrument Resolution," R.J. Anderson and P.R. Griffiths, Anal. Chem., 47, 2339-2347 (1975) and "Extending the

Range of Beer's Law in FT-IR Spectrometry. Part I: Theoretical Study of Norton-Beer Apodization Functions", C. Zhu and P. R. Griffiths, Appl. Spectrosc., 52, 1403-1408 (1998).

A discussion as why the effect of increasing the temperature above ambient and changing the relative humidity reduces the accuracy to which the concentration of the analytes can be predicted would have been useful. I believe that this is caused by the fact that the width of the lines in the vibration-rotation spectra of N2O and CO2 are temperature dependent so that the resolution parameter, , for each line varies with temperature. Similarly, the FWHH of these lines increases with the polarity of the broadening gas, so that it would again be expected that that  would decrease as the temperature increases. The term zero-filling factor (ZFF) appears to be used in different ways depending on which software is being used. For example, I am used to a ZFF of 2 meaning that there are two data points per resolution element. It is stated on page 5 that spectra were calculated with a zero-filling factor of one but Figure 2 shows that there are four data points per resolution element. I would have written that ZFF = 4, not 1. Can the authors state how ZFF is defined in this paper.

The abbreviation S-OPS was not defined in the text (page 10). In the same paragraph it is stated that the physical path length for the OP-FTIR measurements was 150 m, but Figure 1 shows that measurements were also made with physical pathlengths of 50 and 100 m.

These are minor criticisms and I recommend that this excellent manuscript be accepted with only minor changes.
* * *

---

## Referee Comment (RC2) · Anonymous Referee #2 · 24 Aug 2019

The authors describe a study of error sources in a set of innovative measurements of N2O and CO2 emitted from agricultural fields by OP-FTIR and in situ methods (described earlier in a companion paper). The field setup includes in situ measurements from a linear array (50 m long) to characterize the concentration of target gases (N2O and CO2) along an open path of 50 to 150 m. The in situ gas analyzer measurements then serve as a benchmark for quantifying OP-FTIR errors.

The authors consider errors in concentrations derived from FTIR absorption spectra of target gases due to interfering water vapour, atmospheric temperature, open path length as well as the chemometric models used (CLS vs. PLS for N2O, CLS only

for CO2). They also investigate the effects of spectral window width. Finally, they recommend that their quantified error results "be used as references for current or future OP-FTIR users."

General Comments:

The paper is well written and presented, but the description of methodology (the CLS and PLS approach in particular) reads tersely (also in the companion paper and its supplementary material) to someone who uses NLLS for spectral analysis. Furthermore, the discussion of results could drill deeper in places, as outlined below.

The underestimations of N2O referred to in the abstract (from -3% to -30%) were based on concentrations obtained using the CLS model, which assumes a linear relationship between absorbance and target gas concentration. Given the known issues with CLS analysis (e.g., Smith et al., 2011) and the parallel use of the NLLS approach within the OP-FTIR user community, the applicability of these values as references for current or future OP-FTIR users must be qualified accordingly.

P3L17 This is the only spot where NLLS spectral analysis is acknowledged. Four relevant papers are cited but advantages over CLS and PLS are not discussed, even though it takes a lot of work (as outlined in the present paper) to generate the reference libraries central to the CLS and PLS approach. The NLLS approach in the cited works avoids this step entirely by relying on calculated spectra at the correct pressure, temperature and water vapour, and based on spectral parameters from HITRAN. And, as stated by the authors, it overcomes nonlinearity between absorbance and concentration and improves accuracy. What advantage does CLS and PLS offer over NLLS as implemented in the works cited here?

How do the N2O estimations quoted in the abstract reconcile with those from the companion paper (-4.9% for N2O with CLS over 5000-20000 ppmv water from from 10-35C). In general, if there was a discussion of how this study is distinct from the companion paper, which also presents calculated bias results, then I missed it and it needs

to be highlighted more.

Given the innovative field setup where OP-FTIR measurements were made alongside simultaneous gas analyzer measurements drawing air along a linear array, this paper is uniquely positioned to evaluate biases in CLS, PLS and NLLS approaches. This non-trivial effort would greatly increase the relevance of this work to the broadest OP-FTIR community.

Specific Comments:

Section 2.1.2: I found the descriptions of how CLS and PLS models are built and used to derive concentrations from lab absorption spectra (33 m path) unclear, with key method references not sign posted for the un-initiated reader. Why are the N2O reference spectra at 30, 35 and 40 C? Surely these are on the very high end of atmospheric temperatures during an Indiana summer?

Section 2.2.2: I would like to see the the specific rejection criteria used in this study for QA and QC listed, along with the proportion of resultant data loss. Again, not clear on why single-gas reference spectra were generated with HITRAN for CLS while PLS models were built from the lab FTIR measurements. Why not use HITRAN to generate PLS models, too? Also in this section, it is not clear how NLLS regression is used in the CLS model (P6L9) – please explain. Finally, what is the accuracy of the N2O and CO2 gas analyzers that OP-FTIR results are being bench marked against?

Figure 3 shows that water vapour overlaps the N2O P-branch. How does it "compromise" the intensity of the N2O P-branch (P7L21)? The authors suggest that it is via resolution (P7L26), but given how systematic the "compromise is", could it not result from the background correction? Please discuss.

"greater interference at increased temperature" by water vapour (P8L13) presumably means increased line strength in highly temperature-sensitive water vapour lines? Can the worst offenders be avoided via spectral window selection?

Are water vapour and temperature really confounding variables (P8L21) or are their effects in spectra truly indistinguishable (more water vs. greater line strength)? In Figure 5, R2=0.20 (weak) with water and R2=0.86 (strong) with temperature. Furthermore, temperature and RH can be independently measured and in the NLLS approach with calculated HITRAN-based spectra RH and T can be specified independently. Please clarify.

P9L4: In explaining the excess bias in field values of N2O interferences by CO and CO2 are invoked as "presumable". Can one not look at the spectral fit residuals to see if CO and CO2 interferences are being captured correctly?

P9L23: In explaining the short-path bias in field values of N2O, inadequate resolution is invoked as "presumable". Can this not be pinned down more firmly with some test retrievals on synthetic spectra? Is the N2O absorption depth greater than the spectral noise for the 50 m path? Why is the CO2 bias changing at all with path given the very strong absorption signals even at short paths?

P10L14: In explaining the greater bias variability of CO2, the authors presume a greater environmental variation in CO2 than N2O. What would be the biogeochemical and/or physical reason for that? Is respiration (night) more variable than photosynthesis (day)? Do you mean here that 22% of all measurements are calm and at night while 36% of all measurements are calm and during the day? Please clarify.

Technical Correction:

P3L14 single-bean -> single-beam
* * *

---

## Author Comment (AC1) · 5 Oct 2019

The authors thank reviewers for all comments for improving the quality of this paper. More details and deeper discussions suggested by the reviewer will be incorporated in the modified manuscript. The point-to-point responses are as follows: 1. Two citations were suggested a. Errors in absorbance measurements in infrared Fourier transform spectrometry because of limited instrument resolution (R.J. Anderson and P.R. Griffiths, Anal. Chem., 47, 2339-2347 (1975). b. Extending the range of Beer's law in FTIR spectrometry. Part I: Theoretical study of Norton-Beer apodization functions (C. Zhu and P. R. Griffiths, Appl. Spectrosc. 52, 1403-1408, 1998.

[Figure]

2. A discussion as why the effect of increasing the temperature above ambient and changing the relative humidity reduces the accuracy to which the concentration of the analytes can be predicted would have been useful. Response: We did mention the temperature-broadening effect in the manuscript. More discussions will be addressed in the modified manuscript.

3. Define zero-filling factor (ZFF) Response: In this study, the zero-filling factor (ZFF) stood for a power of two (i.e., 2n, n = 1,2, . . ., m), so that 'ZFF of one' meant that one interpolated data point was artificially added within each resolution (0.5 cm-1) to increase the resolution of instrument which was 0.25 cm-1. I think that it would be clearer to express ZFF as two instead of one.

4. S-OPS abbreviation. Response: The term of the S-OPS was defined in the text at the page 6.

5. Clarify the physical paths that we used: not only 150 m, but 50 and 100 m. Response: Three path lengths (physical path = 50-, 100-, and 150-m) were used in the field OP-FTIR experiment. Only the path length of 150-m was used to study the effect of water vapour, temperature, and wind speed on the quantification of N2O and CO2 concentrations in the field. The path lengths of 50-, 100-, and 150-m were used to investigate the effect of the path length on gas quantification. That would be a good idea to simplify the path length from three different paths to one path of 150-m in Figure-1.

---

## Author Comment (AC2) · 5 Oct 2019

The authors appreciate all comments from reviews to improve this paper. Beside more details and deeper discussions that need be incorporated in the modified manuscript, the point-to-point responses are as follows: 1. How do the N2O estimations quoted in the abstract reconcile with those from the companion paper (-4.9% for N2O with CLS over 5,000-20,000 ppmv water from 10-35 °C). In general, if there was discussion of how this study is distinct from the companion paper, which also presents calculated bias results, then I missed it and it needs to be highlighted more.

Response: For both papers, quantitative biases of gas concentrations were calcu-

lated by comparing the path-averaged concentrations between the synthetic open path gas sampling system (S-OPS) and the open-path FTIR (Eq-1 at the P6L31). The main objective of the companion paper (Application of open-path Fourier transform infrared spectroscopy (OP-FTIR) to measure greenhouse gas concentrations from agricultural fields) was to optimize the methods, including the selections of single-bean backgrounds, analytical regions, and multivariate models (CLS vs. PLS), for quantifying gas concentrations. The averaged N2O bias of -4.9±3.1 % was calculated from ninety spectra which contained similar N2O concentrations (338±0.3 ppbv) with different humidity (5,000-20,000 ppmv) and temperature (10-35 °C) using the CLS model. The PLS model was capable of improving the accuracy of gas quantification (i.e. bias =1.4±2.3 %). The main objective of this paper is to evaluate the sensitivities of CLS and PLS models to the ambient temperature and humidity for gas concentration calculations. The results showed that CLS was not only more sensitive to the ambient variables than PLS models for concentration calculations but highly correlated to ambient temperature which likely resulted from the temperature broadening effect of the gas rotation-vibrational absorption features.

2. Section 2.1.2: I found the descriptions of how CLS and PLS models are built and used to derive concentrations from lab absorption spectra (33 m path) unclear, with key method references not sign posted for the un-initiated reader. Why are the N2O reference spectra at 30, 35 and 40 °C? Surely these are on the very high end of atmospheric temperatures during an Indiana summer?

Response: The spectra containing single gas species (e.g. N2O or water vapour only) were used to build CLS models, and the spectra containing mixed-gas species (e.g. N2O mixed with water vapour) were used to build PLS models. Two CLS and one PLS models (CLS-1, CLS-2, and PLS shown in Fig.1) were built based on the source of reference spectra in this study. The CLS-1 model was created using five N2O spectra (i.e. 310, 400, 500, 600, and 700 ppbv) and four water vapour spectra (i.e. 7K, 15K, 22K, and 28K ppmv) collected from the lab-FTIR spectrometer at 30 °C. The CLS-

2 was created using twelve N2O spectra and sixteen water vapor spectra generated from the HITRAN database at 30 °C (see table S1 in the supplement published by Lin et al. 2019). The PLS model was created using a total of sixty mixed-gas spectra (N2O mixed with water vapour) collected from the lab-FTIR spectrometer at 30 °C (see table S1 in the supplement published by Lin et al. 2019). Validation/sample spectra of the wet N2O were both collected from the lab- and OP-FTIR spectrometers to evaluate model performances. The CLS-1 and PLS were used to calculate N2O concentrations from the validation spectra (wet N2O) collected from the lab-FTIR; the CLS-2 and PLS were used to calculate N2O concentrations from the validation spectra collected from the OP-FTIR (Fig.1). Since these quantitative models are temperature-specific (i.e. 30 °C), the temperature variation between reference and the validation/sample spectra leads to biased in gas quantification. For instance, the OP-FTIR validation spectra were collected at 10-35 °C in field experiments (9-19 June 2014), and a strong correlation between temperature and the CLS-quantified biases was observed from the field measurements (Fig-5d in the manuscript). A weak correlation between water vapour and biases, however, were also observed. It was difficult to isolate the effect of the temperature and humidity on quantitative biases from the field experiment, so that the validation spectra with the fixed concentrations of N2O and water vapour (310 ppbv N2O mixed with 21,500 ppmv water vapour) collected at 30, 35, 40 °C from the lab-FTIR to evaluate the sensitivity of CLS and PLS to temperature (Fig.4 in the manuscript). The increased water vapour content was not necessary to increase N2O biases within the water vapour range of 5,000-20,000 ppmv (Fig.2 in the manuscript).

3. Section 2.2.2: I would like to see the specific rejection criteria used in this study for QA and QC listed, along with the proportion of resultant data loss. Again, not clear on why single-gas reference spectra were generated with HITRAN for CLS while PLS models were built from the lab FTIR measurements. Why not use HITRAN to generate PLS models, too? Also in this section, it is not clear how NLLS regression is used in the CLS model (P6L9) – please explain. Finally, what is the accuracy of the N2O and CO2 gas analyzers that OP-FTIR results are being bench marked against?

Response: 1) In this study, there were no specific rejection criteria for ambient temperature and water vapour content, which variation between reference and sample spectra resulted in a quantitative bias. Our study was more interested in the 'delta concentration' between two measurement points to calculate gas fluxes than the absolute concentrations measured from the measuring points. The delta concentration/fluxes were measured every thirty minutes. Since ambient temperature and humidity presumably remained stationary within thirty minutes, the effect of temperature and humidity variation on the delta concentration can be negligible. The path length set-up and wind condition, however, significantly influence the calculation of delta concentrations and fluxes. For instance, a set-up of the short path length (e.g. physical length = 50 m) resulted in greater underestimations than a long path length (e.g. physical length = 100 m or 150 m). Difference path length set-ups (i.e. short and long) likely distorted the actuality of the delta concentration and led to biases in flux estimations. Thus, the criterion for path length is to make sure the same path length used in the measurement system, either short (i.e. physical length less than 100 m) or long (i.e. physical length greater than 100 m) path lengths. The poor-mixed air (wind speed < 2 m s-1) also led to uncertainties in delta concentration, so that the concentrations collected under low wind conditions (< 2 m s-1) were excluded. 2) Ambient variables likely interfere with the quantification of gas concentrations from the OP-FTIR spectra, resulting in an unknown bias. One of the advantages of using PLS models is that numerous ambient variables can be considered in PLS models simultaneously. For instance, reference spectra containing different gas species, concentrations, temperature, humidity, and pathlength, etc. can be considered in one PLS model for concentration calculations. The HITRAN database has been widely used to generate the single-gas spectra to create the quantitative model, mostly CLS and NLLS; however, we had a difficulty to generate the mixed-gas spectra using the HITRAN. Instead, the lab-FTIR was used to collect the spectra which both contained N2O, water vapour, and temperature. 3) The third-degree polynomials were used for the NLLS regression applied in the CLS model using the IMACC software (Industrial Monitoring and Control Corp., Round Rock, TX).

More details are described in the IMACC manual (see the supplement materials). 4) Both N2O analyzer (IRIS 4600, Thermo Fisher Scientific Inc., Waltham, MA) and CO2 analyzer (LI20 840, LI-COR Inc., Lincoln, NE) provided high precision for N2O (< 0.15 ppbv, $1\sigma$) and CO2 (< 1.0 ppmv, $1\sigma$) concentration measurements and calibrated using the certified standard gas every four hour to ensure the stability of analyzers as well as the accuracy for gas measurements.

4. Figure 3 shows that water vapour overlaps the N2O P-branch. How does it "compromise" the intensity of the N2O P-branch (P7L21)? The authors suggest that it is via resolution (P7L26), but given how systematic the "compromise is", could it not result from the background correction? Please discuss.

Response: The N2O (310 ppbv) spectra were acquired in both dry and wet (28,000 ppmv water vapour content) conditions. Ideally, the N2O absorbance/intensity should be identical because of the same concentrations (310 ppbv). The intensity of the N2O P-branch in the wet N2O spectra (red solid line) was observed to be lower than the dry N2O spectra (black solid line) (Figure 3 in the manuscript). To resolve the strong overlap of water vapour in N2O spectra, the inadequate resolution (0.5 cm-1 in this study) was considered as one of the possibilities that cause this issue (intensity reduction in wet conditions). The single-bean backgrounds were acquired before both dry and wet N2O measurements.

5. ''greater interference at increased temperature'' by water vapour (P8L13) presumably means increased line strength in highly temperature-sensitive water vapour lines? Can the worst offenders be avoided via spectral window selection?

Response: Yes, the increased intensity of water vapour with increasing temperature was observed in Figure 4b. It is difficult to avoid the water vapour interference, but this interference could be mitigated via window selection (e.g. WN1 vs. WN3 in CLS models shown in Figure 4a in the manuscript).

6. Are water vapour and temperature really confounding variables (P8L21) or are their

effects in spectra truly indistinguishable (more water vs. greater line strength)? In Figure 5, R2=0.20 (weak) with water and R2=0.86 (strong) with temperature. Furthermore, temperature and RH can be independently measured and in the NLLS approach with calculated HITRAN-based spectra RH and T can be specified independently. Please clarify.

Response: Temperature is considered as a confounding variable influencing both humidity and quantitative bias. The higher temperature tends to have higher water vapour content in the air. In this study, the increased temperature increased biases (R2 = 0.86), and the spurious correlation between water vapour content and bias (R2 = 0.20) likely resulted from temperature effect (i.e. the confounder).

7. In explaining the excess bias in field values of N2O interferences by CO and CO2 are invoked as "presumable". Can one not look at the spectral fit residuals to see if CO and CO2 interferences are being captured correctly?

Response: N2O concentrations were calculated from the analytical window of 2224-2170 cm-1 which includes the information of N2O, water vapour, CO, and CO2 (Fig.2).

8. In explaining the short-path bias in field values of N2O, inadequate resolution is invoked as "presumable". Can this not be pinned down more firmly with some test retrievals on synthetic spectra? Is the N2O absorption depth greater than the spectral noise for the 50 m path? Why is the CO2 bias changing at all with path given the very strong absorption signals even at short paths?

Response: The non-linear relationships between the path length and absorbance respond to different spectral resolution and analyte species (Russwurm and Phillips, 1999: Effects of the nonlinear response of the Fourier-transform infrared open-path instrument on the measurements of some atmospheric gases). For the OP-FTIR spectra, the overlap of multiple species in the spectra further complicated the non-linear responses between path length and absorbance to changing resolution, which might not be easily solved using the single-species synthetic spectra. The physical length of 50-

m/optical path length of 100-m couple with the 64 scans should give us the adequate ratio of single to noise for each spectrum. Compared with the absorption features of N2O at 2170-2224 cm-1, the less complicated features of CO2 absorption make CO2 quantification less sensitive to a short path length than N2O quantification using CLS model (Figure-6 in manuscript). CO2 absorption was also overlapped by water vapour at 2075-2080 cm-1, and the effect of water interferences on gas quantification became severe for the low CO2 absorption spectra acquired from a short path. The 'stronger' absorption signal from a short path was attributed to a strong water vapour signal.

9. P10L14: In explaining the greater bias variability of CO2, the authors presume a greater environmental variation in CO2 than N2O. What would be the biogeochemical and/or physical reason for that? Is respiration (night) more variable than photosynthesis (day)? Do you mean here that 22% of all measurements are calm and at night while 36% of all measurements are calm and during the day? Please clarify.

Response: 1) Since quantitative bias was calculated by comparing the path-averaged concentrations between the S-OPS and OP-FTIR, the spatial distribution of gas concentrations in the atmosphere substantially influence the bias calculation. For instance, CO2 can be produced from both soil and canopy respiration, and plant uptake via photosynthesis. N2O was predominately produced from soil nitrification and denitrification. The CO2 concentrations, as well as their spatial distribution in the air were influenced by the variabilities of both soil properties and crop species (different sources). Thus, CO2 concentrations in the air tended to have higher variabilities than N2O and become highly heterogeneous if the air was poorly mixed in the low wind condition (< 2 m s-1). 2) We do not know if respiration or photosynthesis leads to more variation in CO2 concentrations. 3) During 9-19th 2014, a total of 259 data point (30-min averages) was collected during the daytime measurements (06:00-20:00, LT) and a total of 130 data point was collected from the nighttime measurements (20:00-06:00, LT). The low wind conditions can occur during both day and night (it is more common to have a low wind condition at night). In this study, 22% of all daytime measurements are calm, and 36%

of all nighttime measurements are calm.

| Model | Training/Reference spectra | Validation/Sample spectra |
|-------|---------------------------|---------------------------|
| CLS-1 | **Lab-FTIR:**
N$_2$O spectra: 310, 400, 500, 600, and 700 ppbv
H$_2$O spectra: 7K, 15K, 22K, and 28K ppmv | **Lab-FTIR:**
Wet N$_2$O spectra |
| CLS-2 | **HITRAN database:**
(see Table S1 published by Lin et al, 2019) | **OP-FTIR:**
Wet N$_2$O spectra |
| PLS | **Lab-FTIR:**
A total of sixty wet N$_2$O spectra  (see Table S2 published by Lin et al, 2019) | **Lab- and OP-FTIR:**
Wet N$_2$O spectra |

**Fig. 1.** The information of quantitative models (CLS and PLS), training and validation spectra.

**Fig. 2.** Comparisons between the HITRAN and the OP-FTIR absorption spectra: (a) HITRAN N2O, CO2, CO, and water vapour, and OP-FTIR spectra of 0.338 ppm N2O in (b) low humidity, and (c) high humidity conditions

---

## Author Comment (AC3) · 17 Oct 2019

Dear Referee,

The authors appreciate the work of the nonlinear least squares (NLLS) regressions using in the Multi-Atmospheric Layer Transmission (MALT) model, which is perhaps the most successful method for atmosphere measurements. The authors also understand that the MALT offers numerous advantages, and the most attractive feature to authors is the process of generating single-beam backgrounds, which is one of the sources of errors, is not required for MALT.

Since certain groups of the OP-FTIR users are mainly and still using regular spectral processes and chemometric methods for atmospheric gas measurements, authors only focused on different single-beam background generations, analytical window selections, multivariate regression models (CLS and PLS), and the corresponding errors in this study. Also, one of the purposes of this study is to evaluate the instrument, software, and quantitative methods provided by the United States Environmental Protection Agency in agricultural fields. Thus, the authors did not consider the MALT method in this paper. As mentioned, the companion paper concentrated on the quantitative methodology and this paper focused on the error analysis of these methods. For the CLS non-linearity, the polynomial function was applied in this study to solve the non-linear response of the CLS-calculated concentration to the actual concentration. More details can be found in the supplementary materials.

We do have a unique gas measurement system that can be used for either benchmarking the independent instrumentation or building quantitative models. Authors still engage in optimizing the quantitative methodology for the atmospheric measurement societies. Thus, the authors will consider the MALT method in the future and seek for more collaborations.

Sincerely,

Authors

---

## Editor Decision (ED1)

**Editor's review**

**Sources of error in open-path FTIR measurements of N2O and CO2 emitted from agricultural fields**

**Cheng-Hsien Lin1, Richard H. Grant1, Albert J. Heber2, and Cliff T. Johnston1,3**

This paper is presented as an analysis of systematic errors in quantitative measurements of  $N_2O$  and  $CO_2$  in air using OP-FTIR spectroscopy. As such it can only be useful if the detailed methods used in the analyses are clearly documented in such a way that future practitioners could repeat the analyses on their own data when using the same CLS and PLS techniques. Unfortunately the paper falls short of this requirement, the descriptions in their current form are not sufficiently detailed to be of wider use, and the results pertain only to this particular dataset and analysis. They do provide a useful, qualitative guide to the magnitude of errors that may be encountered.

The authors have adequately responded to some of the referees' comments, but unfortunately not all. In addition and as indicated in my earlier access review I have some editor comments of my own which in many cases overlap those of the referees. I combine these comments below and ask that the authors address these comments before the paper is acceptable for publication in AMT.

Page and line references below are to the "amt-2019-263-author\_response-version2.pdf" document, which includes responses to the two referees' comments.

**Abstract, P8L18-19**

This paper provides a useful qualitative guide to practitioners of OP-FTIR spectroscopy for atmospheric analysis of the systematic errors and biases that arise with commercial low resolution spectrometers using commercial analysis packages based on CLS and PLS chemometric methods. However the authors provide no evidence that the quantitative errors observed and documented in detail here will translate to other instruments and field setups with different resolutions, instrument lineshapes, pathlengths and other conditions. I therefore do not agree with the last sentence of the abstract, which should be removed – this studies serves as a qualitative guide, but not a reference for other users (see also RC2 page C2).

P10L13 (RC2 page C2 ref P3L17) This comment has not been addressed in the authors' response. It is essentially the same comment as made in my initial access review. This work demonstrates significant and complex biases in quantitation using CLS and PLS, including non-linearity and cross-dependency between variables – in this sense it is a useful contribution to the field for the users of commercial chemometric software packages. These sources of error are already well known and recognised from earlier studies. They are less prevalent in a least squares fitting approach to spectrum analysis as now referenced in the papers on L14. RC2's question "What advantage does CLS and PLS offer over NLLS as implemented in the works cited here?" should be addressed, perhaps by a short review paragraph which points out the pros and cons of each approach.

P10L22: "The influences..." The meaning of this sentence is unclear. HITRAN provides a database of absorption line parameters from which an absorption spectrum for any combination of temperature,

pressure and gas composition (including mixtures) can be calculated. This is the approach used in NLLS analysis in which the spectrum is iteratively calculated until a best fit to the measured spectrum is obtained. Please clarify.

P11 section 2.1.1 and P12 section 2.2 – sample and atmospheric pressure:

I cannot find any reference to pressure measurement or control, or pressures used in the CLS/PLS generation of calibration models or analysis of unknowns. Pressure has an important effect on the spectra, as does temperature (for example linewidths are proportional to pressure). In the lab measurements I presume the pressure has been measured and controlled to be the same for calibration and analysis measurements. If so, please state so. But how was pressure included in the open path analysis? Atmospheric pressure will change from hour to hour and day to day, yet the calibration models are presumably built at a single pressure and temperature. Pressure and temperature have two separate effects on retrieved mole fractions in air:

(1) the spectroscopic analysis fundamentally determines a concentration\*pathlength product, from which the concentration (in mol/m3 or similar) is determined. To convert to a mole fraction (eg ppm or ppb) requires the density of air, P/RT. How is this done with the software used?
(2) molecular lineshapes are both pressure and temperature dependent, leading to errors of the calibration spectra and unknown spectra are measured at different pressures and temperatures. This spectroscopic effect is independent of the density effect (1).

The treatment of pressure and pressure variability should therefore be addressed.

P11L17: The synthetic reference mixtures were prepared using N2 as a buffer gas rather than air. Line-broadening coefficients for N2 and air are different, and this will add a systematic bias to the reference measurements of N2O relative to air. This error should be addressed and corrected, or included in the error analysis

P12 L21. Zero filling (RC1): Different FTIR manufacturers and users define "ZFF" differently and it is probably better to avoid this term. From Figure 3 it appears that the point spacing is approx. 0.25 cm-1 or half the 1/maxOPD resolution of 0.5 cm-1. This means the interferogram is minimally sampled and corresponds to NO zero filling, ie the interferogram is extended with zeros only up to the next power of 2 points beyond the highest OPD point (a filling factor of 1, but for example the Bruker and RC1 definition would call this ZFF=2). The best way to resolve this ambiguity is to say "no zero filling" rather than quoting a ZFF value. With no zero filling the point spacing is 0.5/maxOPD.

P11 2.1.2 (RC2) I agree with RC2's comment that the CLS and PLS calibration details are lacking and should be expanded in the paper along the lines in the response – it is not sufficient to respond only to RC2, but to make the methods used clear to all readers in the paper, with sufficient detail that they could repeat the analysis given the same set of data, or their own data. The respective software packages for quantitative analysis have only been presented as a "black box". The claim in the abstract that this paper can be a reference for future work could only stand up if all details of the analysis are presented.

P12 L28: How were the single gas reference spectra calculated from HITRAN data? This is a nontrivial calculation (see eg Griffith 1996), and it is very important to know details of both the molecular spectra and the convolution with the FTIR instrument function if these spectra are to be quantitatively compared with measured spectra (via the CLS/PLS models). It is also not clear exactly what the IMACC software has been used for, or how. In the manual provided in the supplement there is only a description of the user interface, not the underlying calculations or physics behind them. Was IMACC used to calculated SB spectra from Hitran data, or only to generate ABS spectra from measured spectra and background? If not, how were they calculated? None of this is clear. As above, the description should be in principle sufficient for another practitioner to follow your method and achieve the same results.

P13 L16: What is the accuracy (not precision) of the IRIS and Licor analysers used to measure the S\_OPS samples? How were they calibrated. Any systematic bias or error in these calibrations feeds directly into the biases calculated for OP measurements via Eq. 1. (This was answered in a response to RC2, please include in the manuscript)

On the same topic, gas concentrations along the open path will vary with time and location. If the S-OPS measurements are not coincident in time (ie simultaneous) or space (eg along the whole path rather than 150m OP vs 50m S-OPS) an estimate of the potential error needs to be made. At present S-OPS seems to be used as absolute truth without full justification. RC2 points out that this is potentially a very valuable feature of this study, but it cannot be fully exploited without this part of the error analysis.

P14 L26: Increasing resolution will decrease the non-linearity effects of Beer's Law breakdown, but at the cost of signal:noise and hence detection limits and precision. The optimal resolution is a trade-off – this should be included in the discussion here.

P16L18: "Detector saturation at short distances was avoided..." The meaning is unclear. If higher signal levels at short open path distances resulted in saturation (as seen in signal below the detector cut-off), how was this addressed or corrected? Were these spectra rejected? Was the intensity reduced so the detector operated in the linear regime?

**Further comments to RC2**

Section 2.2.2 – please address this point in the manuscript, not only in the response to the referee. The same applies to other comments from RC2 – they need to be addressed in the manuscript.

---

## Author Response (AR2)

**Author's Response**

**Part-I: Point-by-point response**

1. Two citations were suggested
   a. Errors in absorbance measurements in infrared Fourier transform spectrometry because of limited instrument resolution (R.J. Anderson and P.R. Griffiths, Anal. Chem., 47, 2339-2347 (1975).
   b. Extending the range of Beer's law in FTIR spectrometry. Part I: Theoretical study of Norton-Beer apodization functions (C. Zhu and P. R. Griffiths, Appl. Spectrosc. 52, 1403-1408, 1998.

   Response:

   "Zhu and Griffiths, Appl. Spectrosc. 52, 1403-1408, 1998" was included.

2. A discussion as why the effect of increasing the temperature above ambient and changing the relative humidity reduces the accuracy to which the concentration of the analytes can be predicted would have been useful.

   Response:

   We did simply mention temperature-broadening effect in the manuscript. More discussions are addressed in the modified manuscript.

3. Define zero-filling factor (ZFF)

   Response:

   In this study, the zero-filling factor (ZFF) stood for a power of two (i.e., $2^n$, n = 1,2, …, m), so that 'ZFF of one' meant that one interpolated data points was artificially added within each resolution (0.5 $cm^{-1}$) to increase the resolution of instrument which was 0.25 $cm^{-1}$.

4. S-OPS abbreviation.

   Response:

   The term of the S-OPS was defined in the text at the page 6.

5. Clarify the physical paths that we used: not only 150 m, but 50 and 100 m.

   Response:

   Three path lengths (physical path = 50-, 100-, and 150-m) were used in the field OP-FTIR experiment. Only the path length of 150-m was used to study the effect of water vapour, temperature, and the wind speed on the quantification of $N_2O$ and $CO_2$ concentrations in the field. The path lengths of 50-, 100-, and 150-m were used to investigate the

effect of the path length on gas quantification. That would be a good idea to simplify the path length from three different paths to one path of 150-m in Figure-1.

6. How do the $N_2O$ estimations quoted in the abstract reconcile with those from the companion paper (-4.9% for $N_2O$ with CLS over 5,000-20,000 ppmv water from from 10-35 °C). In general, if there was discussion of how this study is distinct from the companion paper, which also presents calculated bias results, then I missed it and it needs to be highlighted more.

Response:

For both papers, quantitative biases of gas concentrations were calculated by comparing the path-averaged concentrations between the synthetic open path gas sampling system (S-OPS) and the open-path FTIR (Eq-1 at the P6L31). The main objective of the companion paper (Application of open-path Fourier transform infrared spectroscopy (OP-FTIR) to measure greenhouse gas concentrations from agricultural fields) was to optimize the methods, including the selections of single-bean backgrounds, analytical regions, and multivariate models (CLS vs PLS), for quantifying gas concentrations. The averaged $N_2O$ bias of -4.9±3.1 % was calculated from ninety spectra which contained similar $N_2O$ concentrations (338±0.3 ppbv) with different humidity (5,000-20,000 ppmv) and temperature (10-35 °C) using the CLS model. The PLS model was capable of improving the accuracy of gas quantification (i.e. bias =1.4±2.3 %). The main objective of this paper is to evaluate the sensitivities of CLS and PLS models to the ambient temperature and humidity for gas concentration calculations. The results showed that CLS was not only more sensitive to the ambient variables than PLS models for concentration calculations, but highly correlated to ambient temperature which likely resulted from the temperature broadening effect of the gas rotation-vibrational absorption features. This study showed that the CLS underestimated N2O concentrations by 3% from lab-FTIR experiment and by 12% from the OP-FTIR experiment when the reference and sample spectra were collected at the same temperature (no temperature variation), which is highlighted in the abstract.

7. Section 2.1.2: I found the descriptions of how CLS and PLS models are built and used to derive concentrations from lab absorption spectra (33 m path) unclear, with key method references not sign posted for the un-initiated reader. Why are the N2O reference spectra at 30, 35 and 40 C? Surely these are on the very high end of atmospheric temperatures during an Indiana summer?

Response:

The spectra containing single gas species (e.g. $N_2O$ or water vapour only) were used to build CLS models, and the spectra containing mixed-gas species (e.g. $N_2O$ mixed with water vapour) were used to build PLS models. Two CLS and one PLS models (CLS-1, CLS-2, and PLS shown in the following table) were built based on the source of reference spectra in this study. The CLS-1 model was created using five $N_2O$ spectra (i.e. 310, 400, 500, 600, and 700 ppbv) and four water vapour spectra (i.e. 7K, 15K, 22K, and 28K ppmv) collected from the lab-FTIR spectrometer at 30 °C. The CLS-2 was created using twelve $N_2O$ spectra and sixteen water vapor spectra generated from HITRAN

database at 30 °C (see the table S1 in the supplement). The PLS model was created using a total of sixty mixed-gas spectra ($N_2O$ mixed with water vapour) collected from the lab-FTIR spectrometer at 30 °C (see the table S1 in the supplement). Validation/wet sample spectra of the wet $N_2O$ were both collected from the lab- and OP-FTIR spectrometers to evaluate model performances. The CLS-1 and PLS were used to calculate $N_2O$ concentrations from the validation spectra (wet $N_2O$) collected from the lab-FTIR; the CLS-2 and PLS were used to calculate $N_2O$ concentrations from the validation spectra collected from the OP-FTIR (see the following table). Since these quantitative models are temperature-specific (i.e. 30 °C), the temperature variation between reference and the validation/sample spectra lead to biased in gas quantification. For instance, the OP-FTIR validation spectra were collected at 10-35 °C in field experiment (9-19 June 2014) and a strong correlation between temperature and the CLS-quantified biases was observed from the field measurements (Fig-5d in the manuscript). A weak correlation between water vapour and biases, however, were also observed. It was difficult to isolate the effect of the temperature and humidity on quantitative biases from the field experiment, so that the validation spectra with the fixed concentrations of $N_2O$ and water vapour (310 ppbv $N_2O$ mixed with 21,500 ppmv water vapour) collected at 30, 35, 40 °C from the lab-FTIR to evaluate the sensitivity of CLS and PLS to temperature (Fig.4 in the manuscript). The increased water vapour content was not necessary to increase $N_2O$ biases within the water vapour range of 5,000-20,000 ppmv (Fig.2 in the manuscript).

**Table1.** Multivariate models (i.e. CLS and PLS) used for $N_2O$ and $CO_2$ quantification: 1) the CLS-1 model was built by five dry $N_2O$ spectra (310, 400, 500, 600, 700 ppbv) and four water vapour spectra (7 000, 15 000, 22 000, 28 000 ppmv) collected from the lab-FTIR, 2) CLS-2 model was built by twelve dry $N_2O$, seventeen dry $CO_2$, and sixteen water vapour spectra generated from the HITRAN database (see Table S1), and 3) the PLS model was built by a total of sixty wet $N_2O$ spectra collected from Lab-FTIR (see Table S2).

| Model | Training/Reference spectra | Source | Validation/Wet Sample spectra | Source |
|---|---|---|---|---|
| CLS-1 | Dry $N_2O$: 310-700 ppbv
Water vaour: 7,000-28,000 ppmv | Lab-FTIR | $N_2O$ | Lab-FTIR |
| CLS-2 | Dry $N_2O$, $CO_2$, water vapour: see Table S1 | HITRAN | $N_2O$, $CO_2$ | OP-FTIR |
| PLS | Wet $N_2O$: see Table S2 | Lab-FTIR | $N_2O$ | Lab- and OP-FTIR |

8. Section 2.2.2: I would like to see the specific rejection criteria used in this study for QA and QC listed, along with the proportion of resultant data loss. Again, not clear on why single-gas reference spectra were generated with HITRAN for CLS while PLS models were built from the lab FTIR measurements. Why not use HITRAN to generate PLS models, too? Also in this section, it is not clear how NLLS regression is used in the CLS model (P6L9) – please explain. Finally, what is the accuracy of the N2O and CO2 gas analyzers that OP-FTIR results are being bench marked against?

Response:

1) In this study, there were no specific rejection criteria for ambient temperature and water vapour content which variation between reference and sample spectra resulted in quantitative bias. Our study was more interested in the 'delta concentration' between two measurement points to calculate gas fluxes than the absolute concentrations measured from the measuring points. The delta concentration/fluxes were measured every thirty minute. Since ambient temperature and humidity presumably remained stationary within thirty minutes, the effect of temperature and humidity variation on the delta concentration can be negligible. The path length set-up and wind condition, however, significantly influence the calculation of delta concentrations and fluxes. For instance, a set-up of the short path length (e.g. physical length = 50 m) resulted in greater underestimations than a long path length (e.g. physical length = 100 m or 150 m). Difference path length set-ups (i.e. short and long) likely distorted the actuality of the delta concentration and led to biases in flux estimations. Thus, the criterion for path length is to make sure the same path length used in the measurement system, either short (i.e. physical length less than 100 m) or long (i.e. physical length greater than 100 m) path lengths. The poor-mixed air (wind speed < 2 m s$^{-1}$) also led to uncertainties in delta concentration, so that the concentrations collected under low wind condition (< 2 m s$^{-1}$) were excluded.

2) Ambient variables likely interfere with the quantification of gas concentrations from the OP-FTIR spectra, resulting in unknown bias. One of the advantages of using PLS models is that numerous ambient variables can be considered in PLS models simultaneously. For instance, reference spectra containing different gas species, concentrations, temperature, humidity, and pathlength, etc. can be considered in one PLS model for concentration calculations. The HITRAN database has been widely used to generate the single-gas spectra to create the quantitative model, mostly CLS and NLLS; so, we only generated the single-gas spectra using the HITRAN. Instead, the lab-FTIR were used to collect the spectra which both contained $N_2O$, water vapour, and temperature.

3) The third-degree polynomials were used for the NLLS regression applied in CLS model using the IMACC software (Industrial Monitoring and Control Corp., Round Rock, TX). More details are described in the IMACC manual (see the supplement materials).

4) Both $N_2O$ analyzer (IRIS 4600, Thermo Fisher Scientific Inc., Waltham, MA) and $CO_2$ analyzer (LI20 840, LI-COR Inc., Lincoln, NE) provided high precision for $N_2O$ (< 0.15 ppbv, 1$\sigma$) and $CO_2$ (< 1.0 ppmv, 1$\sigma$) concentration measurements and calibrated using the certified standard gas every four hour to insure the stability of analyzers as well as the accuracy for gas measurements.

9. Figure 3 shows that water vapour overlaps the N2O P-branch. How does it "compromise" the intensity of the N2O P-branch (P7L21)? The authors suggest that it is via resolution (P7L26), but given how systematic the "compromise is", could it not result from the background correction? Please discuss.

Response:

The $N_2O$ (310 ppbv) spectra were acquired in both dry and wet (28,000 ppmv water vapour content) conditions. Ideally, the $N_2O$ absorbance/intensity should be identical because of the same concentrations (310 ppbv). The intensity of the $N_2O$ P-branch in the wet $N_2O$ spectra (red solid line) was observed to be lower than the dry $N_2O$ spectra (black solid line) (Figure 3 in the manuscript). To resolve the strong overlap of water vapour in $N_2O$ spectra, the inadequate resolution (0.5 cm$^{-1}$ in this study) was considered as one of the possibilities that cause this issue (intensity reduction in wet conditions). The single-bean backgrounds were acquired before both dry and wet $N_2O$ measurements.

10. ''greater interference at increased temperature" by water vapour (P8L13) presumably means increased line strength in highly temperature-sensitive water vapour lines? Can the worst offenders be avoided via spectral window selection?

Response:

Yes, the increased intensity of water vapour with increasing temperature was observed in Figure 4b. It is difficult to avoid the water vapour interference, but this interference could be mitigated via window selection (e.g. $W_N1$ vs. $W_N3$ in CLS models shown in Figure 4a in the manuscript).

11. Are water vapour and temperature really confounding variables (P8L21) or are their effects in spectra truly indistinguishable (more water vs. greater line strength)? In Figure 5, R2=0.20 (weak) with water and R2=0.86 (strong) with temperature. Furthermore, temperature and RH can be independently measured and in the NLLS approach with calculated HITRAN-based spectra RH and T can be specified independently. Please clarify.

Response:

Temperature is considered as a confounding variable influencing both humidity and quantitative bias. The higher temperature tends to have higher water vapour content in the air. In this study, the increased temperature increased biases ($R^2 = 0.86$), and the spurious correlation between water vapour content and bias ($R^2 = 0.20$) likely resulted from temperature effect (i.e. the confounder).

12. In explaining the excess bias in field values of N2O interferences by CO and CO2 are invoked as "presumable". Can one not look at the spectral fit residuals to see if CO and CO2 interferences are being captured correctly?

Response:

$N_2O$ concentrations were calculated from the analytical window of 2224-2170 $cm^{-1}$ which includes the information of $N_2O$, water vapour, CO, and $CO_2$ (see the following Fig.2).

[Figure]

Fig.2 – Comparisons between the HITRAN and the OP-FTIR absorption spectra: (a) HITRAN $N_2O$, $CO_2$, CO, and water vapour, and OP-FTIR spectra of 0.338 ppm $N_2O$ in (b) low humidity, and (c) high humidity conditions.

13. In explaining the short-path bias in field values of N2O, inadequate resolution is invoked as "presumable". Can this not be pinned down more firmly with some test retrievals on synthetic spectra? Is the N2O absorption depth greater than the spectral noise for the 50 m path? Why is the CO2 bias changing at all with path given the very strong absorption signals even at short paths?

Response:

The non-linear relationships between the path length and absorbance respond to different spectral resolution and analyte species (Russwurm and Phillips, 1999: Effects of a nonlinear response of the Fourier-transform infrared open-path instrument on the measurements of some atmospheric gases). For the OP-FTIR spectra, the overlap of multiple species in the spectra further complicated the non-linear responses between path length and absorbance to changing resolution, which might not be easily solved using the single-specie synthetic spectra. The physical length of 50-m/optical path length of 100-m couple with the 64 scans should give us the adequate ratio of single to noise for each spectrum. Compared with the absorption features of $N_2O$ at 2170-2224 $cm^{-1}$, the less complicated features of $CO_2$ absorption make $CO_2$ quantification less sensitive to a short path length than N2O quantification using CLS model (Figure-6 in manuscript). $CO_2$ absorption was also overlapped by water vapour at 2075-2080 $cm^{-1}$, and the effect of water interferences on gas quantification became severe for the low $CO_2$ absorption spectra acquired from a short path. The 'stronger' absorption signal from a short path was attributed to a strong water vapour signal.

14. P10L14: In explaining the greater bias variability of CO2, the authors presume a greater environmental variation in CO2 than N2O. What would be the biogeochemical and/or physical reason for that? Is respiration (night) more variable than photosynthesis (day)? Do you mean here that 22% of all measurements are calm and at night while 36% of all measurements are calm and during the day? Please clarify.

Response:

1) Since quantitative bias was calculated by comparing the path-averaged concentrations between the S-OPS and OP-FTIR, the spatial distribution of gas concentrations in the atmosphere substantially influence the bias calculation. For instance, $CO_2$ can be produced from both soil and canopy respirations, and plant uptake via photosynthesis. $N_2O$ was predominately produced from soil nitrification and denitrification. The $CO_2$ concentrations as well as their spatial distribution in the air were influenced by the variabilities of both soil properties and crop species (different sources). Thus, $CO_2$ concentrations in the air tended to have higher variabilities than $N_2O$ and become highly heterogeneous if the air was poorly mixed in the low wind condition ($< 2$ m s$^{-1}$).
2) We do not know if the respiration or photosynthesis lead to more variation in $CO_2$ concentrations.
3) During 9-19th 2014, a total of 259 data point (30-min averages) was collected during the daytime measurements (06:00-20:00, LT) and a total of 130 data point was collected from the nighttime measurements (20:00-06:00, LT). The low wind conditions can occur during both day and night (it is more common to have a low wind condition at night). In this study, 22% of all daytime measurements are calm, and 36% of all nighttime measurements are calm.

Editor's comments

1. This paper provides a useful qualitative guide to practitioners of OP-FTIR spectroscopy for atmospheric analysis of the systematic errors and biases that arise with commercial low resolution spectrometers using commercial analysis packages based on CLS and PLS chemometric methods. However the authors provide no evidence that the quantitative errors observed and documented in detail here will translate to other instruments and field setups with different resolutions, instrument lineshapes, pathlengths and other conditions. I therefore do not agree with the last sentence of the abstract, which should be removed – this studies serves as a qualitative guide, but not a reference for other users (see also RC2 page C2).

   Response:

   The authors added more details and discussions as the editor and referrers requests in the manuscript. The last sentence in the abstract was rephrased to 'This study identified the most common interferences that affect OP-FTIR measurements of $N_2O$ and $CO_2$, which can serve as a quality assurance/control guide for current or future OP-FTIR users'.

2. P10L13 (RC2 page C2 ref P3L17) This comment has not been addressed in the authors' response. It is essentially the same comment as made in my initial access review. This work demonstrates significant and complex biases in quantitation using CLS and PLS, including non-linearity and cross-dependency between variables – in this sense it is a useful contribution to the field for the users of commercial chemometric software packages. These sources of error are already well known and recognised from earlier studies. They are less prevalent in a least squares fitting approach to spectrum analysis as now referenced in the papers on L14. RC2's question "What advantage does CLS and PLS offer over NLLS as implemented in the works cited here?" should be addressed, perhaps by a short review paragraph which points out the pros and cons of each approach.

   Response:

   A short review paragraph was added in the introduction (the 5th paragraph) to describe the pros and cons of these methods.

3. P10L22: "The influences…" The meaning of this sentence is unclear. HITRAN provides a database of absorption line parameters from which an absorption spectrum for any combination of temperature, pressure and gas composition (including mixtures) can be calculated. This is the approach used in NLLS analysis in which the spectrum is iteratively calculated until a best fit to the measured spectrum is obtained. Please clarify.

   Response:

   The sentence was rephrased and the authors want to emphasize that only limited studies examined how the dynamics of environmental variables affect gas quantification using CLS and PLS models.

4. P11 section 2.1.1 and P12 section 2.2 – sample and atmospheric pressure: I cannot find any reference to pressure measurement or control, or pressures used in the CLS/PLS generation of calibration models or analysis of unknowns. Pressure has an important effect on the spectra, as does temperature (for example linewidths are proportional to pressure). In the lab measurements I presume the pressure has been measured and controlled to be the same for calibration and analysis measurements. If so, please state so. But how was pressure included in the open path analysis? Atmospheric pressure will change from hour to hour and day to day, yet the calibration models are presumably built

at a single pressure and temperature. Pressure and temperature have two separate effects on retrieved mole fractions in air: (1) the spectroscopic analysis fundamentally determines a concentration*pathlength product, from which the concentration (in mol/m3 or similar) is determined. To convert to a mole fraction (eg ppm or ppb) requires the density of air, P/RT. How is this done with the software used? (2) molecular lineshapes are both pressure and temperature dependent, leading to errors of the calibration spectra and unknown spectra are measured at different pressures and temperatures. This spectroscopic effect is independent of the density effect (1). The treatment of pressure and pressure variability should therefore be addressed.

Response:
In lab-FTIR experiment, gas samples were continuously introduced into the White cell with a constant flow rate and the sample pressure was controlled close to the room ambient pressure. The spectra (both training and sample spectra) were collected when the sample concentrations, temperature, humidity and pressure became constant. The training spectra from HITRAN were generated at the pressure of 760 torr. The barometric pressure in the experiment field was measured from 981 to 996 hPa. Compared with the wide ranges of temperature (10-35 °C) and water vapour content (5000 to 20 000 ppmv), the effect of this small variations in the pressure (989.3±3.2 hPa, n = 355) on linewidth as well as gas quantification was not examined and, the measured pressure was only used to adjust the model-calculated concentrations in this study. The potential errors driven by pressure broadening effect was addressed in the results (P10L25). The format of the HITRAN output was specified the product of molar fraction and path length (ppm*m). The HITRAN references were generated at 30 °C and 760 torr. The measured sample/field temperature and pressure can be imported in the quantitative software to adjust the model-calculated molar fraction to the air-density-corrected molar fraction, which was also addressed in P7L4.

5. P11L17: The synthetic reference mixtures were prepared using N2 as a buffer gas rather than air. Line-broadening coefficients for N2 and air are different, and this will add a systematic bias to the reference measurements of N2O relative to air. This error should be addressed and corrected, or included in the error analysis.

Response:
In lab experiment, N2 was used as the background gas to collect both training and sample spectra. Authors specified air as the buffer gas to generate HITRAN training spectra. Therefore, CLS-1 and PLS were based on N2 buffer, and CLS-2 was based on the air. The N2-induced error was addressed in P10L21 to explain the quantitative error of the N2O concentrations calculated from the OP-FTIR spectra using the PLS model.

6. P12 L21. Zero filling (RC1): Different FTIR manufacturers and users define "ZFF" differently and it is probably better to avoid this term. From Figure 3 it appears that the point spacing is approx. 0.25 cm-1 or half the 1/maxOPD resolution of 0.5 cm-1. This means the interferogram is minimally sampled and corresponds to NO zero filling, ie the interferogram is extended with zeros only up to the next power of 2 points beyond the highest OPD point (a filling factor of 1, but for example the Bruker and RC1 definition would call this ZFF=2). The best way to resolve this ambiguity is to say "no zero filling" rather than quoting a ZFF value. With no zero filling the point spacing is 0.5/maxOPD.

Response:
All zero filling was changed to no zero filling to avoid confusion (P6L14).

7.  P11 2.1.2 (RC2) I agree with RC2's comment that the CLS and PLS calibration details are lacking and should be expanded in the paper along the lines in the response – it is not sufficient to respond only to RC2, but to make the methods used clear to all readers in the paper, with sufficient detail that they could repeat the analysis given the same set of data, or their own data. The respective software packages for quantitative analysis have only been presented as a "black box". The claim in the abstract that this paper can be a reference for future work could only stand up if all details of the analysis are presented.

    Response:
    More details were added in 2.1.2 (Data collections and gas quantification for lab-FTIR) and 2.2.2 (Data collections and gas quantification for OP-FTIR) along with the additional table 1. For the details of generating the synthetic background can be found in the companion paper.

8.  How were the single gas reference spectra calculated from HITRAN data? This is a non-trivial calculation (see eg Griffith 1996), and it is very important to know details of both the molecular spectra and the convolution with the FTIR instrument function if these spectra are to be quantitatively compared with measured spectra (via the CLS/PLS models). It is also not clear exactly what the IMACC software has been used for, or how. In the manual provided in the supplement there is only a description of the user interface, not the underlying calculations or physics behind them. Was IMACC used to calculated SB spectra from Hitran data, or only to generate ABS spectra from measured spectra and background? If not, how were they calculated? None of this is clear. As above, the description should be in principle sufficient for another practitioner to follow your method and achieve the same results.

    Response:
    HITRAN: Training absorption spectra used in the CLS model were generated from the HITRAN database using E-trans (Ontar Corporation North Andover, MA). Briefly, high-resolution spectral lines of $N_2O$, $CO_2$, and water vapour output from E-trans were interpolated to generate spectra ranging from 500 $cm^{-1}$ to 4000 $cm^{-1}$ and convolved with a triangular apodization function. The convolved spectral lines were used to generate the reference spectra with the identical resolution and data point density matching the field spectra using Grams/32 (Childers et al., 2001). The HITRAN reference spectra were generated at the pressure of 760 torr and temperature of 30 °C.

    IMACC: The stray-light corrected field SB spectrum was converted to absorbance spectra by the synthetic SB background (syn-bkg) spectra using the IMACC Quantify package (Industrial Monitoring and Control Corp., Round Rock, TX). The syn-bkg was generated by selecting multiple points from the spectral interval of interest (i.e. six points within 2050.0–2500.0 $cm^{-1}$ for $N_2O$ and $CO_2$) to fit the curvature of the sample SB spectrum using a polynomial function (Lin et al., 2019). Three spectral windows (Table 2) and the HITRAN references were used to build the CLS model (CLS-2 shown in Table 1) in the IMACC software. The third-degree polynomial function was used to correct the non-linear response of the CLS-calculated concentration to the actual concentration. More details regarding the IMACC quantification package were described in the IMACC user manual attached as the supplementary material. The PLS models were built using lab-FTIR measurements and only used for estimating N2O concentrations. Since the molar fraction changes with changing air density which is the function of temperature and pressure, the measured temperature and pressure in the gas cell and experimental field were imported in the quantitative software to adjust the model-calculated concentrations.

    Both details were added in the section of 2.2.2 (Data collections and gas quantification for OP-FTIR)

9. P13 L16: What is the accuracy (not precision) of the IRIS and Licor analysers used to measure the S_OPS samples? How were they calibrated. Any systematic bias or error in these calibrations feeds directly into the biases calculated for OP measurements via Eq. 1. (This was answered in a response to RC2, please include in the manuscript)

Response:
Author included in P7L15.

10. On the same topic, gas concentrations along the open path will vary with time and location. If the S-OPS measurements are not coincident in time (ie simultaneous) or space (eg along the whole path rather than 150m OP vs 50m S-OPS) an estimate of the potential error needs to be made. At present S-OPS seems to be used as absolute truth without full justification. RC2 points out that this is potentially a very valuable feature of this study, but it cannot be fully exploited without this part of the error analysis.

Response:
This question has been addressed in author's response in companion paper. Briefly, the path-averaged concentrations of $N_2O$ and $CO_2$ were measured from both S-OPS and OP-FTIR simultaneously, and the S-OPS-measured concentrations were used as benchmarks to examine the accuracy and the sensitivity of OP-FTIR on gas quantification. The surface layer of air tends to become homogeneous in a well mixing condition (i.e. wind speed $> 1.7$ m s$^{-1}$ shown in Lin et al., 2019), and the well-mixed atmospheric condition can minimize spatial variations in the path-averaged concentrations from different measurement units (i.e., 50-m S-OPS vs. 150-m OP-FTIR).

11. P14 L26: Increasing resolution will decrease the non-linearity effects of Beer's Law breakdown, but at the cost of signal:noise and hence detection limits and precision. The optimal resolution is a trade-off – this should be included in the discussion here.

Response:
Authors included this resolution trade-off issue in P9L2.

12. P16L18: "Detector saturation at short distances was avoided…" The meaning is unclear. If higher signal levels at short open path distances resulted in saturation (as seen in signal below the detector cut-off), how was this addressed or corrected? Were these spectra rejected? Was the intensity reduced so the detector operated in the linear regime?

Response:
Detector saturation at short distances was avoided in this study by examinations of the IFG centre burst and SB spectra. For instance, the spectra were excluded if either the maximum or minimum signal of the centre burst exceeded the detector A/DC capacity (~ 2.5 V). Also, the elevated baseline below the detector cut-off, usually 600.0 cm$^{-1}$, in the SB spectrum was used as an indicator to inform the detector saturation (ASTM, 2013).

**Part-II: A list of relevant changes**

Summary – unclear, ambiguous content, and points that needed to be highlighted were rephrased and modified in this manuscript. Also, more discussions and clarifications that were suggested by the reviewers and editors were addressed. The mark-up version if this manuscript was attached (Part-III), and the list of the relevant changes is as follows:

1. **Abstract**

   P1L12 – the CLS underestimated $N_2O$ concentrations by 3% (lab-FTIR) and by 12% (OP-FTIR) experiments when the reference and sample spectra were collected at the same temperature, which is highlighted in the abstract to differentiate the purposes of the companion paper and this article.

   P1L20 – the last sentence in the abstract was rephrased based on the suggestions from the editor and the anonymous referee#2.

2. **Introduction**

   P3L17 – A review paragraph was added in the introduction to describe the strengths and weaknesses these methods.

3. **Materials and experimental methods**

   P5L12 (2.1.2 Data collections and gas quantification for Lab-FTIR) – The entire section was rephrased and one more table (table-1) was added to clarify the methods used in the lab-FTIR experiment which was also suggested by the editor and the anonymous referee#2.

   P6L11 (Data collections and gas quantification for OP-FTIR) – The entire section was rephrased and more information was added to clarify the methods of 1) using the OP-FTIR to acquire spectra, 2) generating HITRAN training spectra, 3) correcting the model-calculated concentrations using the measured temperature and pressure, and 4) calibrating the $N_2O$ and $CO_2$ analysers, which were suggested by the editor and referrers.

4. **Results and discussion**

   Section 3.1.1 Water vapour effect from lab-FTIR experiment
   - P8L21: Explain why the increased water vapour interferences compromised/reduced N2O absorption intensity.
   - P9L2: Increased optimal resolution is a trade-off for the S/N ratio which is along with detection limits as well as quantitative precision.

   Section 3.1.2 Temperature effect from lab-FTIR experiment

   - P9L16: respond to RC2 questions that the water vapour interference could be mitigated by selecting proper spectral windows.

   Section 3.2.1 Water vapour effect from the OP-FTIR experiment

   - P9L24: This section was rephrased to explained how water vapour was confounded by the ambient temperature.

   Section 3.2.2 Temperature effect for OP-FTIR

   - P10L16: the error analysis of the PLS-calculated concentrations was further discussed.

   Section 3.2.3 Wind speed effect

   - P12L7: the multiple sources of $CO_2$ and $N_2O$ were explained to address the variabilities of $CO_2$ and $N_2O$, and the measurements during the low wind conditions were clarified.

   Section 3.2.3 Path length effect

   - Avoiding detector saturation was addressed in P11L10.

   Section 3.2.4 Wind speed effect

- More variations in CO2 than N2O concentrations were explained in P12L7, and the data collected from the low wind conditions were clarified in P12L12.

[revised manuscript text omitted]

---

## Author Response (AR3)

**Author's Response to Associate Editor for Minor Comments**

**Part-I: Point-by-point response**

Notes: The page and line numbers in editor's comments refer to Author_response_v3.pdf. The page and line numbers in responses refer to this author response version.

1. Throughout – ppmv and ppbv are not concentrations (ie amount per volume), they are mole fractions. It is OK to use "concentration" in a general sense, but not when referring to specific ppm or ppb values – please use "mole fraction" in these (many) cases throughout.

   Response:

   The 'concentrations' referring to the specific ppm/ppb have been changed to mole fraction throughout this manuscript.

2. P14L23 – CO2 is also a significant GHG from agriculture.

   Response:

   $CO_2$ was added in P4L23.

3. P16L6 omit " OP-FTIR measurements … instrument . Consequently"

   Response:

   This part was removed.

4. P16 L24 MALT is just one example of the NLLS/forward modelling approach, you could also mention the forward model you have used, E-trans from ONTAR Corp. (Note this was not mentioned in earlier versions of the MS, but is now specified P19 L20).

   Response:

   The E-trans software developed by the Ontar Corp. was added in this section at P6L23.

5. P16 L32 Griffith 2009 seems to be missing from the references.

   Response:

   This is a typo. It was supposed to be Griffiths 2009, which was fixed.

6. P18L4. Given that this is a quantitative study it is important to specify ACCURACY of reference measurements. Here, what is the accuracy of the diluter? How accurate are the figure od 30ppmv, 310.0, 400.0, 500.0, 600.0, 700.0 ppbv? Is the nnn.0 significant figure really correct??

Response:

The accuracy of the Environics 4040 diluter was ±1.0 % for gas mole fraction measurements, which was addressed in the method section at P9L2, so the decimal point of the diluted $N_2O$ were removed (P8L2).

7. P19 L20. To be clear with the ambiguity of specifying zero filling, I suggest adding …no zero filling (ie spectral datapoint spacing ~ 0.25 cm-1).

Response:

The zero-filling factor was specified as no zero filling with the spectral datapoint spacing ~ 0.5 $cm^{-1}$ at P9L14.

8. P19 L20. How often were the stray light spectra measured (they may change with time and temperature). P20L11. Could you please clarify if for the OP spectra you have used IMACC software to build and use the CLS and PLS models, rather than TQ Analyst as used for the lab spectra. Why use two different packages?

Response:

The stray light spectrum was acquired every thirty minutes, which was addressed at P9L16. The IMACC software only allowed users to build the CLS models, so the TQ Analyst was used to build the PLS model for gas quantification, which was clarified in the method section of 2.2.2 at P10L2

9. P20L30. As above, please specify the certified accuracy (not precision) of the reference CO2 and N2O measurements.

Response:

The $N_2O$ and $CO_2$ calibration gas was certified using EPA protocol (1% certification), which was addressed at P10L18.

10. P24 L19. How were the reference mole fractions for the path length study determined? Are they from the S-OPS over 50 m? Please be explicit, also with the accuracy of the assumption that the S-OPS values can be extrapolated over the longer 100 and 150 m paths.

Response:

The reference mole fractions were measured from the 50-m S-OPS. When the atmosphere is in a well-mixed condition, which typically occurs at the wind speed greater than 2 m $s^{-1}$, the surface layer as well as gas concentrations tend to become invariant under a spatial translation (i.e. homogeneous). Therefore, the path-averaged mole fractions measured from the 50-, 100-, or 150-m S-OPS were presumably invariant in this field scale (< 150 m). In this study, we only used the 50-m S-OPS to measure the ambient gas concentrations which were used as reference $N_2O$ and $CO_2$

mole fractions to assess the OP-FTIR path length effect on gas quantification. This assumption was addressed in both sections of 2.2.2 (P10L21) and 2.2.3 (P11L1).

**Part-II: A list of relevant changes**

Summary – a further clarification and minor changes were made based on comments from the editor. The mark-up version of this manuscript was attached (Part-III), and the list of the relevant changes is as follows:

1. **Abstract**
   Changed 'concentration' to 'mole fractions' in this section.
   P4L11 – rephrased.
2. **Introduction**
   Several sentences had minor modifications in this section.
   P6L23 – E-trans software, one of the forward modelling approaches, was added in this section.
3. **Materials and experimental methods**
   Most of the 'concentrations' were rephrased to 'mole fractions' in the method section.
   P8L2 (2.1.1 Instrumentation setup) – the accuracy ($\pm 1.0$ %) of the diluter for mole fractions was added in this section.
   P9L14 (Data collections and gas quantification for OP-FTIR) – no zero-filling (i.e. spectral datapoint spacing ~ 0.25 cm$^{-1}$) and the frequency for stray-light SB spectra collections were added.
   P10L2 – The quantitative models (CLS and PLS) and their corresponding software (IMACC and TQ Analyst) for the OP-FTIR spectra analysis were clarified.
   P10L18 – the accuracy of the certified gas ($\pm 1.0$ %) for both $N_2O$ and $CO_2$ analysers was addressed in this section.
   P11L1 – we clarified the reason why the path-averaged mole fraction measured from the 50-m S-OPS can be used as the reference mole fraction to assess the performance of the OP-FTIR measurements from different physical path of 100- and 150-m in a well-mix atmospheric condition (the surface layer becomes homogenous).
4. **Results and discussion**
   Several sentences in this section had minor modifications.

[revised manuscript text omitted]